# The SC-35 Splicing Factor Interacts with RNA Pol II and A-Type Lamin Depletion Weakens This Interaction

**DOI:** 10.3390/cells10020297

**Published:** 2021-02-01

**Authors:** Soňa Legartová, Paolo Fagherazzi, Lenka Stixová, Aleš Kovařík, Ivan Raška, Eva Bártová

**Affiliations:** 1Institute of Biophysics, Academy of Sciences of the Czech Republic, Královopolská 135, 612 65 Brno, Czech Republic; legartova@ibp.cz (S.L.); fagher@ibp.cz (P.F.); lenka@ibp.cz (L.S.); kovarik@ibp.cz (A.K.); 2Department of Experimental Biology, Faculty of Science, Masaryk University, Kamenice 753/5, 625 00 Brno, Czech Republic; 31st Faculty of Medicine, Charles University, Albertov 4, 128 00 Praha, Czech Republic; iraska@lf1.cuni.cz

**Keywords:** splicing, SC-35, PARP inhibitor, RNA pol II

## Abstract

The essential components of splicing are the splicing factors accumulated in nuclear speckles; thus, we studied how DNA damaging agents and A-type lamin depletion affect the properties of these regions, positive on the SC-35 protein. We observed that inhibitor of PARP (*poly (ADP-ribose) polymerase*), and more pronouncedly inhibitors of RNA polymerases, caused DNA damage and increased the SC-35 protein level. Interestingly, nuclear blebs, induced by PARP inhibitor and observed in A-type lamin-depleted or senescent cells, were positive on both the SC-35 protein and another component of the spliceosome, SRRM2. In the interphase cell nuclei, SC-35 interacted with the phosphorylated form of RNAP II, which was A-type lamin-dependent. In mitotic cells, especially in telophase, the SC-35 protein formed a well-visible ring in the cytoplasmic fraction and colocalized with β-catenin, associated with the plasma membrane. The antibody against the SRRM2 protein showed that nuclear speckles are already established in the cytoplasm of the late telophase and at the stage of early cytokinesis. In addition, we observed the occurrence of splicing factors in the nuclear blebs and micronuclei, which are also sites of both transcription and splicing. This conclusion supports the fact that splicing proceeds transcriptionally. According to our data, this process is A-type lamin-dependent. Lamin depletion also reduces the interaction between SC-35 and β-catenin in mitotic cells.

## 1. Introduction

It is well known that processes including 5′-capping, 3′-polyadenylation, and RNA splicing lead to the functional mRNA formation that is the final product of precursor-mRNA (pre-mRNA) processing. Transcription is a process in which nascent RNA is produced according to the DNA template. A primary transcript must undergo RNA processing, including the above-mentioned 5’ capping and polyadenylation (an appearance of the poly-A tail). Finally, constitutive splicing is responsible for the withdrawal of intron sequences, considered as non-protein-coding regions. Before terminal splicing, both ends of a pre-mRNA must be modified at the 5’ end. There is a structure called a cap in this region, and the 3’ end represents the so-called tail structure. The cap consists of modified guanine (G) nucleotides that protect the transcript. The polyadenylation signal is responsible for the transcript stability, which is essential for mRNA transport to the cytoplasm. The crucial components of the splicing machinery are also small nuclear ribonucleoproteins (snRNPs) [1]. Notably, a serine/arginine-rich splicing factor 2 (SC-35, encoded by the *SRSF2* gene) also contributes to the splicing processes. The mentioned components of the cell nucleus are accumulated in so-called nuclear speckles, which are considered highly dynamic structures of the cell nucleus [2]. Nuclear speckles are arranged into 20–50 accumulated foci associated with interchromatin granule clusters (IGCs) [3,4,5]. Notably, during mitosis, especially in the metaphase, nuclear speckles are disassembled, while in the telophase, splicing factors form so-called mitotic interchromatin granules (MIGs) [6]. In MIGs, the SC-35 protein is hyperphosphorylated [7].

Nevertheless, when we disregard alternative splicing, producing distinct mRNAs from one gene, constitutive splicing works in two specific ways: first, so-called *cis*-splicing that appears within an identical pre-mRNA molecule. However, *trans*-splicing works on two separate pre-mRNA molecules to create the non-co-linear RNA that encodes proteins or can serve as non-coding RNA, having a direct regulatory function. Splicing in *trans* can proceed in both prokaryotes and eukaryotes [8,9,10]. In human cells, *trans*-splicing of pre-mRNA appears in many physiological and pathological processes, including cell transformation into a malignant genotype [11,12]. Thus, genome instability in tumor cells leads to the subsequent synthesis of aberrant proteins, which are functionally distinct from their physiological counterparts. This decisive process of mRNA translation into fully functional or aberrant polypeptides proceeds in ribosomes, components of the rough endoplasmic reticulum (ER) that, together with smooth ER, are located in the cytoplasm. It is well known that the ER represents more than half of the eukaryotic cell’s membranous components [13].

In the cells, alternative splicing of pre-mRNA is also a very specific nuclear event. This process contributes to gene variability and functioning, even though only one gene encodes distinct mRNA, and subsequently, several protein products. This nuclear phenomenon is essential for the development of the organism and tissue-specific cell differentiation. It is well known that alternative splicing is regulated by many factors, like *cis*-acting and *trans*-acting elements. To these molecular biology processes, structural features, including the cell’s nuclear architecture, contribute to both constitutive and alternative splicing [14].

Similarly, epigenetic factors like DNA methylation and post-translational modifications of histones regulate splicing processes [15,16]. Moreover, microRNAs and small interfering RNAs are considered as regulators of alternative splicing [17]. Thus, the function of non-coding RNAs in splicing machinery should also be studied from the view of epitranscriptomic features, including RNA post-transcription modification, such as N^6^-adenosine methylation (m^6^A). The m^6^A in distinct RNAs is thought to be an essential signal for pre-mRNA splicing initiation [18]. It is well known that constitutive splicing mediates intron removal and exon ligation, while alternative splicing enables, for instance, retention of introns or exon shuffling [19]. An example of the alternatively spliced gene is the *lmna* gene encoding both lamin A and lamin C, components of the nuclear lamina, in which disorders lead to laminopathy diseases [20].

The above-mentioned processes and molecular regulations inspired us to study cellular localization and dynamic properties of the main splicing factor, SC-35. Due to the fact that Ilik et al. [21] showed that the antibody against SC-35 instead determines the SRRM2 protein, we verified, in a subset of experiments, the data by other antibodies, e.g., the antibody against SRRM2, and the SRSF7 protein as well. Primarily, by immunofluorescence, we studied the properties of the SC-35 protein by the use of anti-SC-35. In this regard, we analyzed the effect of damaging agents, including poly (ADP-ribose) polymerase (PARP) inhibition, RNA pol I or II inhibitors, γ-irradiation, and A-type lamin depletion. It is well known that PARP inhibition causes DNA damage, accompanied by phosphorylation of histone H2AX (γH2AX) [22].

Similarly, in lamin A/C-deficient cell populations, there are more cells with delayed disappearance of γH2AX foci and DNA repair factor recruitment. Thus, A-type lamins’ physiological function is essential for maintaining genome stability after DNA damage [23]. From this viewpoint, we also studied the PCNA protein that regulates not only DNA replication but also DNA damage repair [24,25]. As an additional DNA repair marker, we studied the UBF1 protein, the transcription factor for ribosomal genes, which is also a component of the nucleotide excision repair (NER) mechanism [26].

As our results showed that SC-35 is decorating the plasma membrane of mitotic cells, we studied a link between the SC-35 protein and another protein associated with the plasma membrane and regulating both cell-to-cell adhesion and gene expression. For that reason, we analyzed β-catenin [27]. Together, we studied how the distribution of nuclear speckles and interaction properties of the SC-35 splicing factor is affected by DNA damaging agents and A-type lamin deficiency.

## 2. Materials and Methods

### 2.1. Cell Cultivation and Treatment

The human cervix adenocarcinoma (HeLa) cell line (ATCC^®^ CCL-2^TM^) was cultivated in EMEM (Eagle’s minimum essential medium, Merck, Germany) supplemented with 10% fetal calf serum (FCS) and the appropriate antibiotics. Mouse embryonic fibroblasts (MEFs), wild-type (wt), and A-type lamin-depleted (double knockout; dn) was a generous gift from the team of prof. Collin Stewart [28], and were cultivated in DMEM (Dulbecco’s modified Eagle’s medium, Merck, Germany) with 10% fetal calf serum and appropriate antibiotics [29]. The following human cell lines were also studied: A549, HL60, MCF7, MOLP8, U2OS, U937, HaCaT, and IMR-90 cells. The cultivation conditions of the cell lines mentioned above are summarized in Table 1. The human colorectal adenocarcinoma (HT29, #ATCC^®^ HTB-38™, Germany) cells were cultivated in McCoy’s 5A Medium (#M9309, Merck, Germany) supplemented with fetal calf serum to a final concentration of 10%. The HT29 cells were treated with the inhibitor of histone deacetylases (HDACs), sodium butyrate (NaBt; final concentration 5 mM; #B5887, Sigma Aldrich, Czech Republic), inducing differentiation in enterocytes (48-h treatment was used; see Bartova et al. [30]). The human embryonic stem cells (hESCs, line CCTL-017, a derivative cell line of CCTL-12, a gift from the Department of Biology, Faculty of Medicine, Masaryk University, Brno, Czech Republic) were cultured in mTESR1 medium (#05870, Stemcell Technologies, USA) on Matrigel-coated plates (#354277, Corning^®^Matrigel^®^ hESC-Qualified matrix, Corning, NY, USA). To induce differentiation, hESCs colonies were treated with 2 mM all-trans-Retinoic acid (ATRA, #R2625, Merck, Darmstadt, Germany), and the cells were harvested after a 2-day differentiation [31].

Human embryonic stem cells (CCTL-017) were maintained according to Czech national law 227/2006 and the Ethics Committee agreement No.616/2012/31. The cells were maintained at 37 °C in a humidified atmosphere containing 5% CO_2_, and for experiments, cells were harvested at approximately 80% confluence. The HeLa cells were treated with PARP inhibitor (PARPi) Olaparib (#S1060, Selleckchem, München, Germany) with a final concentration 10 μM, for 24 h or RNA pol II inhibitor α-amanitin (#A2263, Merck, Germany), final concentration 2 μg/mL for 2 h (for FRAP experiments, we additionally used a 24 h-treatment). Cells were also treated with 0.5 μg/mL actinomycin D (ActD, #A9415, Merck, Germany) for 2 h.

For irradiation by γ-rays, cells were exposed to 5 Gy of γ-rays, delivered by cobalt-60 (Chirana, Prague, Czech Republic). Thirty minutes after γ-irradiation, cells were either fixed for immunostaining or harvested for western blots. We also studied the effect of γ-radiation and its combination with PARPi treatment (cells were harvested 24 h after the treatment; with a dose 5 Gy of γ-rays, cells were harvested 30 min after irradiation).

For transfection, the cells were cultivated on 35-mm glass-bottom tissue culture dishes (#81158, µ-Dish 35 mm, high, Ibidi, Gräfelfing, Germany) to 70% confluence and transfected with 2–5 μg plasmid DNA encoding mCherry-tagged PCNA (originally from the laboratory of prof. Christina Cardoso, Technical University, Darmstadt, Germany) and GFP-tagged UBF1/2 (#17656, Addgene, Watertown, MA, USA; [38]). Transfection was performed using METAFECTENE^®^PRO reagent (no. T040-2.0, Biontex Laboratories GmbH, München, Germany), and concentrations were optimized for individual plasmids. For plasmid amplification, we used chemically competent E. coli DH5α bacteria, and plasmid DNA was isolated by using a QIAGEN Plasmid Maxi Kit (#121693, Qiagen, Bio-Consult, Prague, Czech Republic).

### 2.2. Immunofluorescence Staining

Immunofluorescence was modified, following Bartova et al. [39]. The cells were fixed in 4% formaldehyde (PFA) for 10 min at room temperature (RT), permeabilized with 0.2% Triton X-100 (#194854, MP Biomedicals, Santa Ana, CA, USA) for 8 min, and 0.1% saponin (#S7900, Sigma Aldrich, Czech Republic) for 12 min. After that, the dishes were washed twice in phosphate buffer saline for 15 min. We used 1% bovine serum albumin (BSA; #A2153-506, Sigma Aldrich, Prague, Czech Republic) dissolved in 1× PBS, used as a blocking solution, in which the samples were incubated for one hour at room temperature and then washed in 1× PBS for 15 min. For immunofluorescence analysis, the following antibodies were used: anti-SC-35 (#ab11826, Abcam, Bristol, UK), anti-SRSF7 (#HPA056926, Merck, Germany), anti-SRRM2 (#HPA041411, Merck, Germany), anti-lamin A (#ab26300, Abcam, UK), β-catenin (#06-734, Sigma Aldrich, Czech Republic), and anti-RNA polymerase II, CTD repeat YSPTSPS (phospho S5; Pol II pS5) (#ab5131, Abcam, UK). The following secondary antibodies were used: Alexa Fluor 488-conjugated donkey anti-mouse (#A21202, Thermo Fisher Scientific, Waltham, MA, USA), Alexa Fluor 594-conjugated goat anti-mouse (#A11032, Thermo Fisher Scientific, USA), Alexa Fluor 488-conjugated goat anti-rabbit (#ab150077, Abcam, UK), and Alexa Fluor 594-conjugated goat anti-rabbit (#A11037, ThermoFisher Scientific, USA). Primary antibodies were not used in the samples tested as a negative control. A contour of cell nuclei (condensed chromatin; [40]) was visualized by the use of 4′,6-diamidino-2-phenylindole (DAPI; #D9542, Merck, Germany), dissolved in Vectashield (#H-1000, Vector Laboratories, Burlingame, CA, USA).

### 2.3. Western Blot

Western blots were performed following Legartova et al. [41]. Cell cultures or tissue samples were washed with PBS and lysed in sodium dodecyl sulfate (SDS) lysis buffer (50 × 10^3^ mol/L Tris-HCl, pH 7.5; 1% SDS; 10% glycerol). The total protein concentration was determined by a DC protein assay kit (#5000111, Bio-Rad, Prague, Czech Republic) and ELISA Reader μQuant (BioTek, Winooski, VT, USA). The proteins were separated by SDS polyacrylamide gel electrophoresis (SDS-PAGE) and transferred to polyvinylidene difluoride (PVDF) membranes. The membranes were blocked with 2% nonfat milk or 2% gelatin for one hour, and then immunoblotted overnight at 4 °C with the following primary antibodies: anti-SC-35 (#A12625, ABclonal, Woburn, MA, USA), anti-SRSF7 (#HPA056926, Merck, Germany), anti-SRRM2 (#SAB2108778, Merck, Germany), anti-γH2AX (#ab2893, Abcam, UK), anti-β-catenin (#06-734, Sigma Aldrich, Czech Republic), anti-lamin A (#ab26300, Abcam, UK), and anti-PARP1 (#A2432, ABclonal, USA). Primary antibodies were diluted 1:1000. Note, the antibody against the SRRM2 protein (C-terminal) recognizes the following epitope: DKKEKSATRPSPSPERSSTGPEPPAPTPLLAERHGGSPQPLATTPLSQEP (34 kDa). As secondary antibodies, we used goat anti-rabbit IgG (#AP307P, Merck, Czech Republic; 1:2000), anti-mouse IgG (#A9044, Sigma-Aldrich, Czech Republic; 1:2000), and goat anti-mouse IgG1 (#ab97240, Abcam, UK). The western blot membranes were stained with amino acid staining azo dye (amido black) to detect the transferred membrane blots’ total protein level. The western blot data were normalized to the total protein level. The normalization to total protein level considers all of the proteins in the sample, and their total abundance serves for data normalization. The quantification itself was performed using ImageJ software (NIH freeware) as following; first, we selected the area of identical size around the western blot fragment. Then, ImageJ software produced histograms (one histogram per fragment) with numerical values that were further assessed.

### 2.4. Laser Scanning Confocal Microscopy

We acquired images with a Leica TCS SP8X SMD confocal microscope (Leica Microsystem, Wetzlar, Germany), equipped with 63× oil objective (HCX PL APO, lambda blue) with a numerical aperture (NA) 1.4. Image acquisition was performed using a white light laser (WLL; wavelengths of 470–670 nm in 1-nm increments) with the following parameters: 1024 × 1024-pixel resolution, 400 Hz, bidirectional mode, and zoom 8. For 3D projections, we acquired 30–40 optical sections with an axial step of 0.2 μm. Reconstruction of 3D projection was done using the Leica Application Suite (LAS X) software, and the Leica Lightning software performed the deconvolution procedure.

### 2.5. FLIM-FRET Technique

Fluorescence lifetime image microscopy (FLIM) combined with Förster resonance energy transfer (FRET) was performed, following Legartova et al. [42]. Using this method, we studied the interaction between SC-35 protein (donor) and β-catenin (acceptor), as well as EGFP-SRSF2 (donor) and A-type lamin (acceptor). Additionally, we studied the link between SC-35 (donor) and RNA polymerase II (acceptor). For the detection of the protein-protein interaction, we mainly used fixed samples, immunostained with the following antibodies: anti-SC-35 (#ab11826, Abcam, UK), anti-RNA polymerase II CTD repeat YSPTSPS (phospho S5; Pol II pS5) (#ab5131, Abcam, UK), and anti-β-catenin (#06-734, Sigma Aldrich, Czech Republic). As secondary antibodies we used: goat anti-mouse Cy3 (#ab97035, Abcam, UK) and goat anti-rabbit Cy5 (#ab6564, Abcam, UK).

To detect EGFP-SRSF2 and A-type lamin interaction, cells were first transiently transfected with plasmid DNA encoding EGFP-tagged SRSF2 (the SRSF2-EGFP N1 plasmid DNA was a gift from Dr. Béatrice Eymin, Institute for Advanced Biosciences, Grenoble, France, [43]). After that, immunostaining was performed by the following antibodies: anti-lamin A (#ab26300, Abcam, UK) or anti-RNA polymerase II CTD repeat YSPTSPS (phospho S5; Pol II pS5) (#ab5131, Abcam, UK). As a secondary antibody, we used Alexa Fluor 594-conjugated goat anti-rabbit (#A11037, Thermo Fisher Scientific, USA). Proteins SC-35/EGFP-SRSF2 (donor) and RNA polymerase II (pS5; acceptor) were considered as the well-interacting partners for FRET experiments.

The fluorophore characteristics used for FRET experiments were adopted from the webpage https://www.fpbase.org/fret/ (see Table 2).

Fluorophores were characterized according to their absorption and fluorescence properties, such as molar extinction coefficient (EC) or quantum yield (QY). The EC measures the ability of a fluorescence molecule to absorb light. A higher extinction coefficient leads to absorbing a greater amount of light [44]. In our FRET experiments, we used two different sets of FRET pairs: Cy3 (donor)/Cy5 (acceptor) and EGFP (donor)/Alexa Fluor 594 (acceptor). The molar extinction coefficients for Cy3 (donor)/Cy5 (acceptor) were 2.7 times higher in comparison to EGFP (donor)/Alexa Fluor 594 (acceptor). Spectra overlapping integral J(λ) were significantly higher for the Cy3/Cy5 fluorophore pair.

This means that even though we used similar well-interacting partners, we obtained different FRET efficiencies. The FRET efficiency was higher for SC-35 stained with Cy3, and RNA polymerase II pS5 stained with Cy5, than EGFP-tagged SRSF2 and lamin A stained with Alexa Fluor 594. These differences can be explained by the distinct physical properties of the distinct fluorochromes. According to our experience, 25% of FRET efficiency can be considered as the cut-off level indicating protein-protein interaction. Thus, E-values ≥25% show significant interaction.

Samples were mounted in Vectashield (#H-1000, Vector Laboratories, Burlingame, CA, USA). Measurement was performed by the use of a Leica TCS SP8 X SMD confocal microscope (Leica Microsystems GmbH, Wetzlar, Germany), a PicoHarp 300 module (PicoQuant GmbH, Berlin, Germany), and HyD SMD detectors. For cell visualization, we used a 63× oil immersion objective of numerical aperture 1.4. Results were analyzed by SymPhoTime 64 software (PicoQuant GmbH, Germany), and FRET efficiency was calculated following [45,46].

### 2.6. FRAP Analysis

HeLa cells were cultivated on 35-mm dishes with a glass-bottom (µ-Dish 35 mm, high, Ibidi) and transfected with EGFP-tagged SRSF2 [43]. For FRAP analysis, cells on cultivation dishes were placed into a cultivation hood (EMBL, Heidelberg, Germany), maintained at 37 °C and 5% CO_2_. Time-lapse imaging of live cells was carried out in a confocal Leica TSC SP8 microscope, equipped with an argon laser and objective with 64× magnification, and the numerical aperture NA = 1.4. The time of pre-bleaching was 2.61 s, during which fluorescence intensity (FI) was measured. Bleaching of defined regions (2 μm^2^) was performed using an argon laser at 100% laser power (laser line 488 was used). Bleaching was performed for 0.783 s, and post-bleaching observation for 20.88 s, with an interval of 0.261 s. During the whole procedure, the fluorescence intensity was measured. Scanning was done at 0.5% power of the 488-nm argon laser and with a frame resolution of 512 × 512 pixels. The scanning rate was 1000 Hz, and an 8x zoom was used. We analyzed data using LEICA LAS AF lite 4 software, and line graphs were created by Excel software.

### 2.7. Statistical Analysis

Data were analyzed using the Mann–Whitney U test (STATISTICA software), a non-parametric test of the null hypothesis applied for X and Y values, randomly selected from two experimental units. We also used regression analysis, available in Sigma Plot software, version 14.0 (Jandel Scientific, San Rafael, CA, USA). Pearson’s correlation coefficient was calculated via Leica LAS X software and its colocalization tool.

## 3. Results

### 3.1. The SC-35 Protein Was Localized in Nuclear Blebs, and Inhibitors of RNA Polymerases Changed Its Level

The SC-35 protein was localized in 52.2 ± 3.5% of the nuclear radius in HeLa cells, and the size of the nuclear speckles was approximately 4.79 ± 0.78 µm^3^ (Figure 1a–c and Figure 2a). SC-35 occupied interchromatin space (IC), characterized by a low DAPI density, due to a low chromatin density in these cell nucleus regions. In HeLa cells, treated with PARP inhibitor olaparib, we observed the presence of nuclear blebs and micronuclei, as in A-type lamin-depleted (dn) cells, as well as in senescent A-type lamin wild type (wt) mouse embryonic fibroblasts (50th passage) (Figure 2b,c and Figure 3a,b). These nuclear blebs were SC-35 positive in all cases studied (Figure 2b,c and Figure 3a,b). Interestingly, in A-type lamin-depleted cells, SC-35 positive foci inside the cell nuclei were more diffused in comparison to lmna wt fibroblasts (Figure 3a,b).

Since the treatment by PARP inhibitor, olaparib, induced the appearance of nuclear blebs and micronuclei (Figure 2b,c), we analyzed if apoptosis appears in these cases. Western blots showed no apoptotic fragmentation of lamin A and PARP1; therefore, nuclear blebbing was not an effect of an additional nuclear event, apoptosis (Figure 4a). Therefore, the appearance of non-physiological nuclear structures, like nuclear blebs, should be a consequence of lamin disruption and abrogation of lamin interaction with heterochromatin at the nuclear periphery, which might also be caused by PARP inhibition, additionally inducing pronounced DNA damage (see an increased level of phosphorylation of histone H2AX (γH2AX) in PARPi treated cells Figure 4a).

In addition, we studied the effect of γ-irradiation on the nuclear distribution of the SC-35 protein. Irradiation by γ-rays did not affect the SC-35 localization in the cell nucleus (Figure 2b) and caused its slight depletion, observed by western blots (Figure 4b). We also analyzed how α-amanitin treatment can change the localization and the level of the SC-35 protein. Both immunofluorescence and western blotting showed that the SC-35 protein pool increased in cells treated with RNA pol II inhibitor, the α-amanitin (Figure 2d,e and Figure 4b). Similarly, an inhibitor of RNA polymerase I and II, actinomycin D (ActD), also enhanced the SC-35 protein level (Figure 4b). However, the antibody against SRRM2 did not show these changes. Only the SRSF7 protein was up-regulated in the cells treated with ActD, decreasing the level of β-catenin (Figure 4a,b). Except for α-amanitin, all treatments caused DNA damage, characterized by pronounced γH2AX positivity (Figure 4a).

As additional experiments, we performed FRAP analysis showing that 24 h after α-amanitin treatment, the recovery time after photobleaching of EGFP-tagged SRSF2 is more pronounced in comparison to non-treated cells (Figure 4c). Other treatments, including γ-irradiation, PARPi, and their combination, did not affect EGFP-SRSF2 recovery time after photobleaching (Figure 4c). This observation fits well with the western blot data, also showing changes in the SC-35 protein properties when the cells were treated with α-amanitin (Figure 4a,b). Interestingly, PARPi up-regulated SC-35, while its combination with γ-irradiation, maintained the SC-35 level comparable with the control values (Figure 4b). These data seem to be valuable from the perspective of potential clinical applications.

### 3.2. The Highest Level of SC-35 Was Accompanied by Depletion of A-Type Lamins in Distinct Cell Types

Here, we also provide data on the SC-35 protein level studied by western blot in various normal and tumor cell types; we found distinct, cell-type-specific levels of the SC-35 protein. For instance, human embryonic stem cells (hESCs) and human leukemia cell lines (HL-60 and U937) were characterized by the highest level of the SC-35 protein, which was accompanied by a deficiency of A-type lamins (Figure 5a). We also studied a correlation between the levels of SC-35 and lamin A, or SC-35 and β-catenin. Results of regression analysis are shown. The correlation coefficient was r^2^ = 0.0056 for SC-35/lamin A, and r^2^ = 0.0097 for SC-35/β-catenin (n − 2 = 11) (Figure 5b). These results are not statistically significant, but there was the following trend in western blots: the highest levels of SC-35 were accompanied by barely detectable lamin A (see Figure 5a, arrows and asterisks). This rule was observed for pluripotent hESCs, their partially differentiated counterparts, and non-differentiated progenitor leukemia cells HL60 and U937 (Figure 5a).

### 3.3. A Spatial Link of SC-35 Positive Nuclear Speckles to DNA Repair Proteins and Nucleoli

With regard to the DNA repair process, we additionally analyzed whether the SC-35 protein directly colocalizes with 53BP1-positive DNA repair foci in the non-irradiated control HeLa cells (spontaneously occurring DNA repair foci were studied) and cells irradiated by 5Gy of γ-rays, inducing so-called irradiation-induced foci (IRIF) (Figure 6a,b). In the previous experiments with MEFs [48], we instead found an association between SC-35-positive nuclear speckles and 53BP1-positive DNA repair foci. For an explanation, these foci did not appear to colocalize directly but were in close proximity. This was observed for mostly spontaneously occurring DNA repair foci in mouse embryonic fibroblasts (MEFs) [48]. Moreover, transmission electron microscopy showed a high positivity of the 53BP1 protein in MEF cells’ nuclear speckles [48]. However, we observed a random association between the SC-35 protein and 53BP1-positive DNA repair foci in human tumor HeLa cells. These regions were relatively mutually disconnected, as was shown by confocal microscopy combined with an image deconvolution software tool (Figure 6a,b). According to these data, HeLa tumor cells are characterized by the distinct nuclear distribution of the SC-35 protein compared to MEFs (Figure 6a,b, and [48]). In the subsequent analyses, we studied the spatial distribution of the SC-35 protein in the compartment of the nucleoli, visualized according to GFP-tagged UBF1 protein, concerned as a transcription factor for the ribosomal genes, and also found to be a factor of the nucleotide excision repair mechanism (Figure 7a,b; [26]). We did not observe colocalization between SC-35 and the GFP-tagged UBF1 protein, but SC-35 positive nuclear speckles decorated nucleoli, as is well visible in Figure 7b. It was reported by Bubulya et al. [49] that nuclear speckles in the G1 phase accumulate around nucleolar organizing regions (NORs), forming nucleoli, and these structures were established as NOR-associated patches (NAP).

### 3.4. The SC-35 Protein Decorates the Plasma Membrane in Mitotic Cells and the Degree of Its Colocalization with PCNA Is Enhanced in the Late S-phase of the Cell Cycle

In mitosis, tiny foci of the SC-35 protein were positioned on the cell periphery. The abundance of SC-35 in the cytoplasm of the telophase cells was also published by Tripathi and Parnaik [7]. However, in HeLa cells, we observed a very pronounced ring of endogenous SC-35 protein decorating the cell periphery; thus, SC-35 seems to be associated with the plasma membrane (Figure 8a). Moreover, we addressed the question of if the SC-35 protein has a link to lamin proteins that surrounded mitotic chromosomes (Figure 8b). In this regard, Tripathi and Parnaik [7] showed that GFP-SC-35 in telophase colocalized with the nascent nuclear envelope. Here, we found that while the SC-35 protein is associated with the plasma membrane, A-type lamins are mostly decorated metaphase rosettes (Figure 8b). Due to the fact that Ilik et al. [21] showed that the monoclonal antibody against SC-35 preferentially recognizes the SRRM2 protein, but not SRSF2, we additionally studied distribution properties by the use of additional antibodies against SRRM2, and also SRSF7.

In comparison to anti-SC-35, both antibodies showed a distinct distribution pattern of the proteins studied in mitosis. In detail, anti-SRRM2 and anti-SRSF7 did not occupy the cell periphery. Interestingly, in telophase and early cytokinesis, SRRM2 formed robust cytoplasm foci that resemble MIGs. As expected, in the interphase, SRRM2 was accumulated in nuclear speckles and micronuclei, but SRSF7 was homogeneously dispersed through the nucleoplasm of the interphase cells (Figure 8e,f). The plasmid encoding EGFP-SRSF2 showed a high density of exogenous SRSF2 in the cytoplasm of mitotic cells; the highest GFP-SRSF2 positivity was between metaphase rosettes (Figure 8g).

Here, we also studied the link of SC-35 to proliferation marker PCNA that plays a role in DNA replication and DNA repair. In this case, we observed a relatively high Pearson’s correlation coefficient (0.41) in non-S cells, and this parameter was enhanced in the late S-phase of the cell cycle (0.66) (Figure 8c,d).

### 3.5. A-Type Lamin-Dependent Interaction between the SC-35 Protein and RNA Polymerase II in Interphase Cells, or SC-35 and β-catenin Interaction in Mitosis

By using FLIM–FRET analysis, in interphase cell nuclei, we showed a significant interaction between the SC-35 protein and the phosphorylated form of RNAP II (FRET efficiency was 75.0% ± 2.7%). In mitotic cells, the SC-35 protein interacted with β-catenin (FRET efficiency was 36.5% ± 3.0%) (Figure 9a,c). β-catenin was selected as a potential interacting partner with SC-35 in mitosis due to its localization on the plasma membrane, which was also decorated by SC-35 (Figure 9d,e). In this regard, the depletion of A-type lamins weakened both SC-35/RNAP II and SC-35/β-catenin mutual interactions (FRET efficiency was reduced to 23.2% ± 5.2% and 20.3% ± 4.2%, respectively); (Figure 9b,d). In mitotic cells, with normal function of A-type lamins, SC-35 co-localizes with β-catenin at the periphery of the cells (plasma membrane), while A-type lamin depletion diminished the SC-35 positivity in peripheral rings, as well as colocalization between SC-35 and β-catenin (Figure 9e). Quantification of the fluorescence intensity of visualized proteins showed a rearrangement of the SC-35 protein in mitotic cells when we compared lmna wt and lmna dn fibroblasts (see quantification in Figure 9f). In A-type lamin-depleted cells, SC-35 was rather dispersed in the cytoplasm, and was not accumulated at the cell periphery, as was observed in lmna wt fibroblasts (Figure 9e,f). Analysis of the average fluorescence intensity of the SC-35 protein in mitotic cells showed a different distribution of the SC-35 fluorescence intensities in the cytoplasm of lmna dn cells and their wild type counterparts (Figure 9f). An interaction between splicing factor and RNAP II was verified by the use of EGFP-tagged SRSF2 protein, showing a relatively high degree of interaction with the phosphorylated form of RNAP II in both wt MEFs and HeLa cells (FRET efficiency was 27–29%) (Figure 9g(i,ii)). When we analyzed the interaction between EGFP-tagged SRSF2 and lamin A, FRET efficiency was relatively low (~18%) (note, E ≥ 25% is considered as the FRET efficiency showing protein–protein interaction, see Materials and Methods section) (Figure 9g(iii,iv)).

## 4. Discussion

It is well known that nuclear speckles are membrane-less structures located in so-called interchromatin space, and they are considered as the sites of the splicing process [4,50]. Moreover, Galganski et al. [51] suggested that nuclear speckles’ function is also linked to the transcription process. From this viewpoint, we studied properties of the main splicing factor, the SC-35 protein, in cells exposed to DNA damaging agents, including clinically used PARP inhibitor and γ-radiation. The physiological function of A-type lamins is essential for maintaining genome stability after DNA damage, and internal lamins protect the replication fork; thus, we also addressed how A-type lamin depletion affects the SC-35 protein properties.

At this point, we must point out that Ilik et al. [21] showed that monoclonal antibody against nuclear speckles, specified here as anti-SC-35, preferentially recognizes the SRRM2 protein, but not SRSF2 (syn. SC-35), as it was previously published [52]. Ilik et al. [21] also showed that the SON factor, together with SRRM2, is responsible for nuclear speckle formation. Importantly, here, we observe an identical morphology of nuclear speckles recognized by anti-SC-35 and anti-SRRM2 in interphase, but the morphology was different in mitotic cells (Figure 1b and Figure 8a,e,g). By western blots, we also did not see identical results with anti-SC-35 or anti-SRRM2, but anti-SC-35 showed relatively similar results to anti-SRSF7, indicating an increased level of the studied proteins in the cells treated by α-amanitin and Act-D (Figure 4b). This observation validates the findings of Ilik et al. [21].

We approach this topic from the fact that in the past, most studies were done by the use of anti-SC-35; thus, we continued to perform our analysis with this antibody. We found that PARP inhibitor, olaparib, significantly affected the level of splicing factor SC-35 (Figure 4b). We observed a similar phenomenon in the cells treated with an RNA polymerase II inhibitor, α-amanitin, and an inhibitor of RNA polymerase II/I, actinomycin D (Figure 4b). It is well known that, for example, α-amanitin inhibits RNA polymerase II, an enzyme responsible for the transcription of protein-coding genes, so an increase in the pool of the SC-35 splicing factor was surprising. Similarly, the same trend we observed in the cell treated by PARPi, so this fact should be considered when evaluating the PARP inhibition effect on splicing, especially of tumor suppressor genes, including BRCA1. In this case, a crucial observation is that γ-irradiation combined with PARPi maintained the SC-35 level comparable with the control values (Figure 4b).

Interestingly, olaparib additionally induced the formation of nuclear blebs and micronuclei that appear to be functional due to SC-35 positivity (Figure 2b,c). This observation implies that pre-mRNA splicing occurs co-transcriptionally, even in non-physiological structures like nuclear blebs [53]. We also confirmed co-transcriptional regulation of splicing by our FLIM-FRET results, showing a high degree of interaction between SC-35 and the phosphorylated form of RNAP II (Figure 9a,b). Similarly, Shimi et al. [54] showed a high density of hyperphosphorylated RNA polymerase II and histone marks associated with active transcription in nuclear blebs that, according to this observation, can also be considered as reservoirs of transcription factors. However, in vivo, RNA labeling showed that transcription was decreased in these non-physiological structures appearing on the periphery of cell nuclei [54]. Furthermore, Bercht Pfleghaar et al. [55] confirmed that gene-rich chromosomal regions are enriched in nuclear blebs, and there is an active form of RNA polymerase II. So, these lamin A/C-rich but lamin B-poor structures [54] seem to be, at least, reservoirs of the stalling of RNA polymerase II and the SC-35 protein (the SRRM2 protein, respectively) (Figure 2c, Figure 3a,b and Figure 8e). According to Funkhouser et al. [56], nuclear blebs represent aberrant gene expression regions, especially in laminopathy cells. In blebs, the lamin meshwork has a more significant separation; thus, lamin-associated meshwork is in a more open configuration, affecting the regulation of transcription and pre-mRNA splicing [56]. The link between the level of A-type lamins and splicing machinery we explained by the following results: Although regression analysis did not show a correlation between levels of SC-35 and lamin A in distinct cell types (Figure 5b), western blot analysis documents the following rule: an absence of lamin A in the cells was accompanied by the highest levels of SC-35 protein (Figure 5a, arrows, and asterisks). This phenomenon was observed in pluripotent hESC cells and progenitor HL60 and U937 leukemia cells. We found the same trend when we observed the reduced fluorescence intensity of SC-35 in the peripheral ring of lmna dn cells compared to the wild type counterpart (Figure 9f). Based on these data, we summarized that there is a functional link between the SC-35 protein and A-type lamins in the cells, maintaining genome stability.

Moreover, both internal and peripheral lamins stabilize not only nuclear speckles but the entire nuclear contents [57]. Thus, the appearance of nuclear blebs should be the consequence of lamin disruption, which leads to the abrogation of the heterochromatin interaction with the nuclear lamina. This effect might be caused by PARP inhibitor, as observed here (Figure 2b,c). According to some authors, cells in the S-phase of the cell cycle are more prone to nuclear budding and blebbing. Morphological findings of these abnormal nuclear structures appear in parallel with the p53 and pKi-67 overexpression associated with the faulty mitotic checkpoint or mitotic catastrophe, especially in cancer cells [58,59].

Besides a high positivity of SC-35 in nuclear blebs, we also observed the SC-35 protein’s localization on the mitotic cells’ periphery (Figure 8a). Tripathi and Parnaik [7] showed the positioning of SC-35 in the nascent nuclear envelope of telophase cells. They documented, using the FRAP technique, that the mobility of GFP-tagged SC-35 was different in various mitotic compartments. For instance, the mobility of GFP-SC-35 was significantly higher in the cytoplasm of metaphase cells compared with speckles appearing in the interphase cell nuclei. Here, we observed by FRAP that α-amanitin potentiates EGFP–SRSF2 fluorescence recovery after photobleaching, studied in interphase cells (Figure 4c). The presence of SC-35 or SRRM2-positive foci at the cytoplasm of mitotic cells show that these foci represent reservoirs of splicing factors in mitosis. In this regard, Galganski et al. [51] documented that nuclear speckles disassemble during the early stages of mitosis and reassemble in telophase, which seems to be the case for the SC-35-positive peripheral ring and SRRM2-positive foci in the late telophase (Figure 8a,e). Similarly, Rai et al. [60] showed SRRM2–mCherry granule formation in mitotic cells, treated with GSK-626616 compound, an inhibitor of the dual-specificity kinase DYRK3. These authors also showed a high density of the SC-35 protein associated with the plasma membrane of the cells in telophase.

Taken together, the data shown here confirm the existence of well-defined structures of the splicing machinery in both interphase and the late telophase. Still, a question remains of whether nuclear speckles are anchored into steady compartments or if there is stochastic self-assembly of factors associated with nuclear speckles, as suggested by Dundr and Misteli [61]. According to all the mentioned data, it is evident that the formation of speckle-like structures is established during late mitosis in the cytoplasm, containing a reservoir of splicing factors that are arranged into nuclear speckles in the interphase.

## Figures and Tables

**Figure 1 cells-10-00297-f001:**
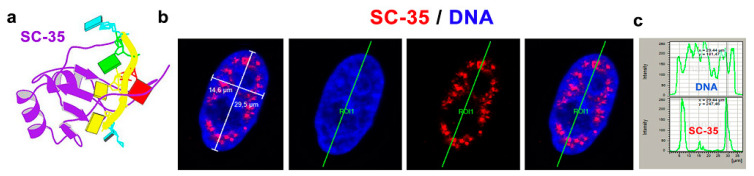
Structure and nuclear distribution of the SC-35 protein. (**a**) The structure of SC-35 protein in a complex with 5′-UCCAGU-3′ motif was adapted from the PDB database (deposition authors were Daubner et al. [47]). (**b**) The SC-35 protein (red fluorescence) was localized in 52.2 ± 3.5% of the nuclear radius (measured in 50 cells in each experiment, repeated thrice). (**c**) Analysis of fluorescence intensity showed that SC-35 occupies sites with a low chromatin density, as visualized by the 4′,6-diamidino-2-phenylindole (DAPI) staining (blue fluorescence).

**Figure 2 cells-10-00297-f002:**
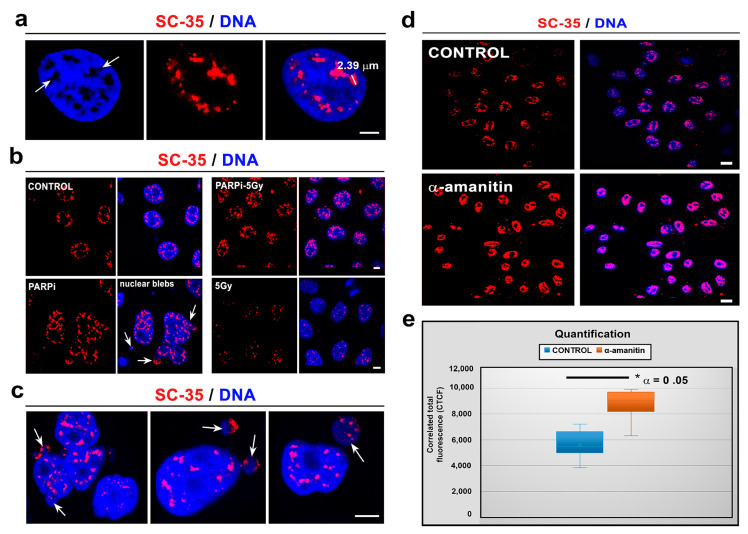
The localization of SC-35 in HeLa cells treated with potential DNA damaging agents like *poly (ADP-ribose) polymerase* (PARP) inhibitor, RNA pol inhibitors, and γ-radiation. (**a**) SC-35 occupies interchromatin space. DAPI was used for visualization of the low density of chromatin. The selected nuclear speckle diameter was 2.38 µm; the volume of nuclear speckles was 4.79 ± 0.78 µm^3^. (**b**) Nuclear distribution of SC-35 in non-treated HeLa cells, and cells treated with a PARP inhibitor, olaparib, and 5Gy of γ-rays. (**c**) A detail of nuclear blebs and micronuclei, positive on the SC-35 protein (red fluorescence signal). Nuclear blebs and micronuclei mainly appeared after the cell treatment by PARP inhibitor, olaparib. White arrows show nuclear blebs and micronuclei. (**d**) Nuclear distribution of SC-35 in non-treated HeLa cells and cells treated with RNA pol II inhibitor, α-amanitin. Bars indicate 10 µm in panels (**a**–**d**). (**e**) The chart shows the corrected total fluorescence value (CTCF) for each cell analyzed using ImageJ software; the following formula was used: CTCF = integrated density (area of selected cells x mean fluorescence of background readings). A total number of fifty cells were used for analysis. Quantification was done in non-over saturated images; for image presentation and printing, the saturation was increased. The non-parametric Mann–Whitney test was used for statistical analysis. Significance is shown for * α = 0.05.

**Figure 3 cells-10-00297-f003:**
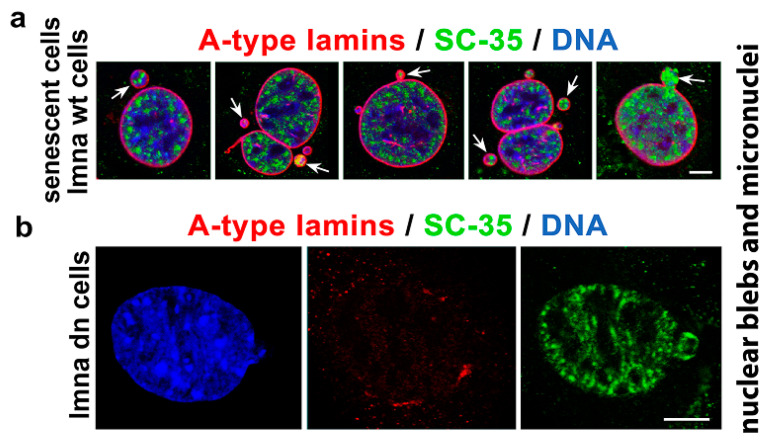
The localization of SC-35 in nuclear blebs of senescent lmna wt and lmna dn cells. (**a**) The SC-35 protein (green) occupied nuclear blebs and micronuclei of the cells with normal A-type lamin function but senescent (red, wt cells). (**b**) Similarly, cells with depletion of A-type lamins (lmna dn cells) were also characterized by SC-35 positive nuclear blebs. DAPI was used for visualization of chromatin density in the cell nuclei (blue). White arrows show nuclear blebs or micronuclei. Bars represent 10 µm.

**Figure 4 cells-10-00297-f004:**
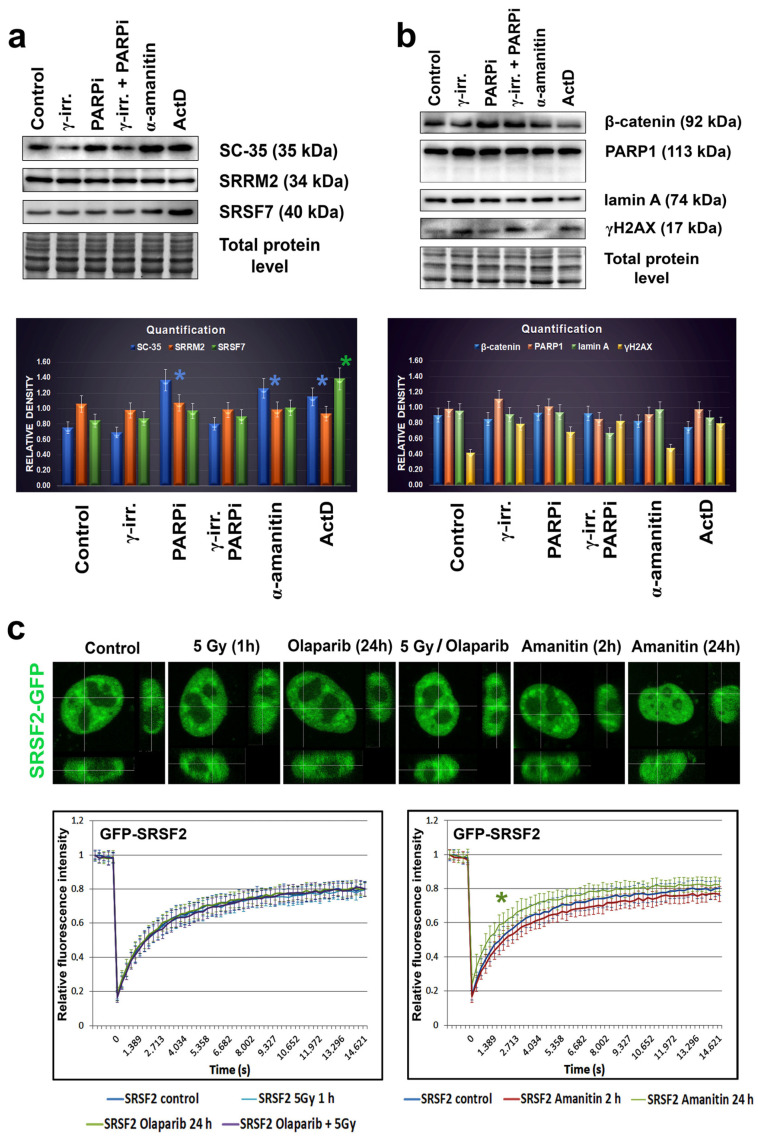
Selected protein levels in HeLa cells treated with PARP or RNA polymerases’ inhibitors and irradiated with γ-rays. Panels (**a**,**b**) The following proteins’ levels were studied: PARP1, γH2AX, lamin A, β-catenin SC-35, SRRM2, and SRSF7. Western blots showed no apoptotic fragmentation of lamin A and PARP1. Quantification of western blot data was performed by ImageJ software. The density of western blot fragments was normalized to the pool of total proteins. Asterisks show increased protein levels. The non-parametric Mann–Whitney test was used for statistical analysis. Asterisks (*) mean α = 0.05. (**c**) FRAP analysis showed EGFP-SRSF2 recovery kinetics after photobleaching performed in non-treated HeLa cells, and the cell population irradiated with γ-rays, treated with PARPi (olaparib) or affected with α-amanitin. The green asterisk shows the most rapid fluorescence recovery after photobleaching of the EGFP-tagged SRSF2 protein in the cells treated with α-amanitin for 24 h. Data were compared with the control values.

**Figure 5 cells-10-00297-f005:**
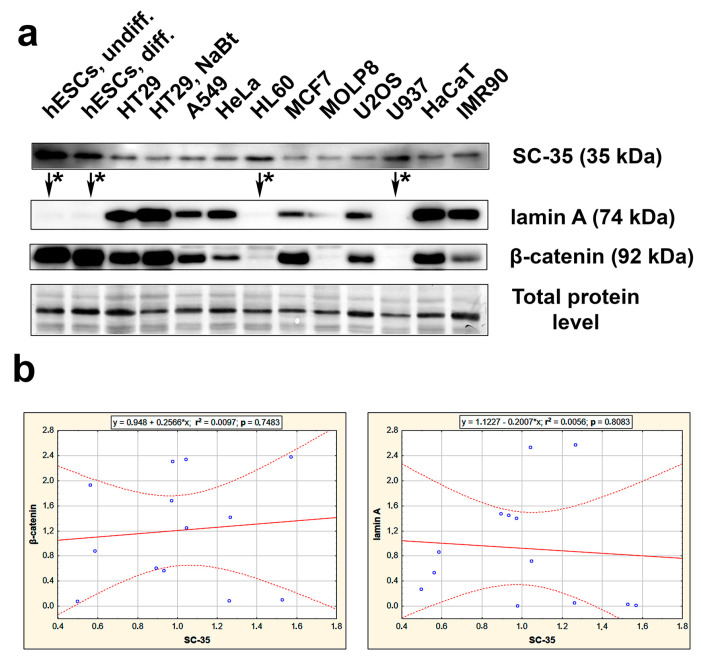
Studies on SC-35, lamin A, and β-catenin in distinct cell types. (**a**) The level of selected proteins was studied in the following cells: human ES cells (hESCs), partially differentiated hESCs, tumor cell lines, including HT29, A549, HeLa, HL60, MCF7, MOLP8, U2OS, U937, and human keratinocytes HaCaT, and fibroblasts IMR90. Data on protein levels were normalized to the total protein levels. In cell lines studied, the highest SC-35 levels were accompanied by barely detectable lamin A (see arrows and asterisks). (**b**) Regression analysis, evaluating the correlation between SC-35 and lamin A (*p* = 0.8083; r^2^ = 0.0056), or SC-35 and β-catenin (*p* = 0.7483; r^2^ = 0.0097), is shown.

**Figure 6 cells-10-00297-f006:**
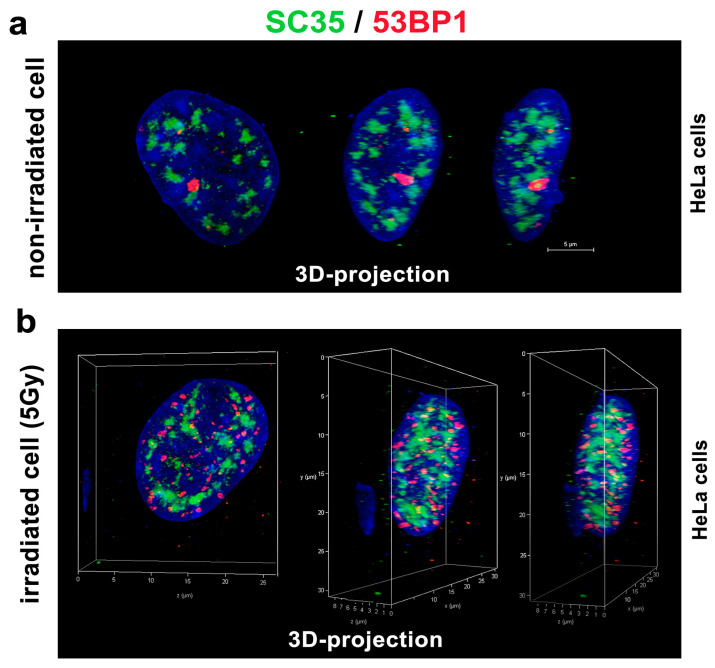
The 3D projection of SC-35 protein-positive foci and the DNA repair protein 53BP1 in HeLa cells’ nucleus. Nuclear distribution and mutual distance between SC-35 and 53BP1 proteins in (**a**) non-irradiated control cells, and (**b**) cells irradiated by 5 Gy of γ-rays. The deconvolution procedure was applied to these images. The 3D reconstruction of confocal images was performed by Leica LAS X software. Scale bar shows 5 µm.

**Figure 7 cells-10-00297-f007:**
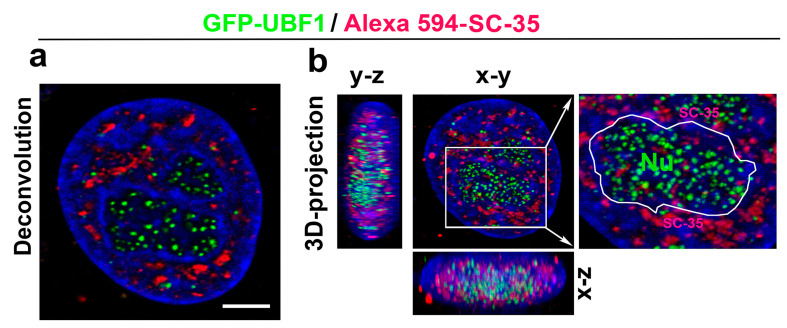
The spatial link between the SC-35 protein and the compartment of nucleoli. Nucleoli were visualized by GFP-tagged UBF1 protein (green), also playing a role in the nucleotide excision repair (NER) repair mechanism. Nuclear speckles were recognized according to the Alexa 594-stained SC-35 protein (red). (**a**) HeLa cell nuclei were visualized by conventional confocal and deconvolution tool. Scale bar shows 5 µm. (**b**) 3D-projection of confocal images is shown. The compartment of the nucleolus (Nu), decorated by the SC-35 protein, is enlarged. The nucleolus boundary was depicted according to the UBF-1 positivity and location of DAPI-dense chromatin (white contours).

**Figure 8 cells-10-00297-f008:**
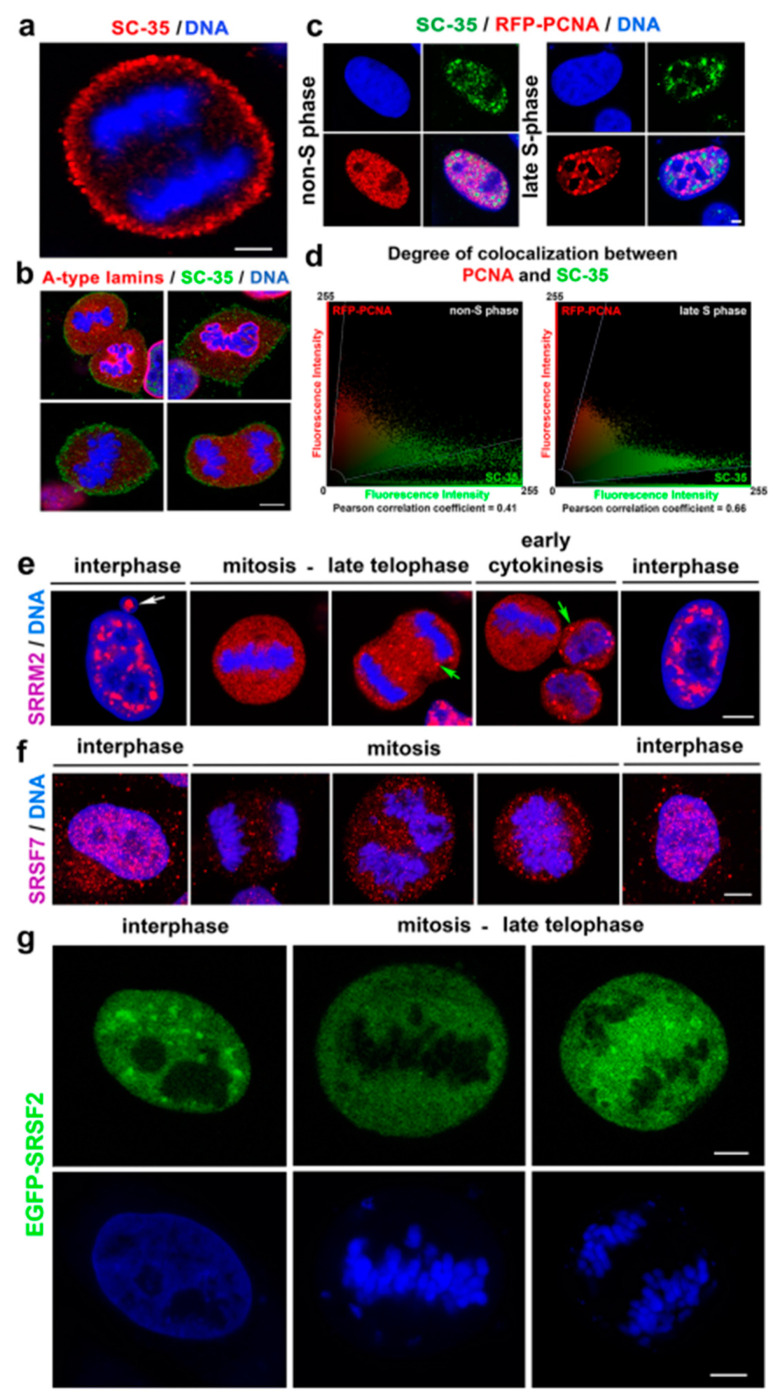
Nuclear localization of the following proteins: SC-35, PCNA, SRRM2, and SRSF7. (**a**) The SC-35 protein accumulated in tiny foci, localized in proximity to the plasma membrane. A high density of EGFP-SRSF2 was found in the cytoplasm of the cells in telophase. The analysis was performed in HeLa cells. Scale bar shows 7 µm. (**b**) No colocalization was observed between SC-35 and A-type lamins in mitotic HeLa cells. Scale bar shows 10 µm. (**c**) A link of SC-35 (green) to RFP-tagged PCNA (red) in both non-S phase cells and the cells in the late S-phase of the cell cycle. (**d**) Pearson’s correlation coefficient for SC-35 and PCNA was 0.41 in non-S cells, and this coefficient was higher (0.66) in the cells in the late S-phase; see an example in this panel showing analysis of cells in panel (**c**). Localization of (**e**) SRRM2 and (**f**) SRSF7 proteins in the interphase, mitosis-late telophase, and early cytokinesis. The white arrow shows nuclear bleb, and the green arrows show SRRM2-positive foci in the cytoplasm of late telophase and at the early cytokinesis stage. (**g**) Distribution profile of GFP-tagged SRSF2 in interphase and mitotic HeLa cells. Scale bar shows 5 µm.

**Figure 9 cells-10-00297-f009:**
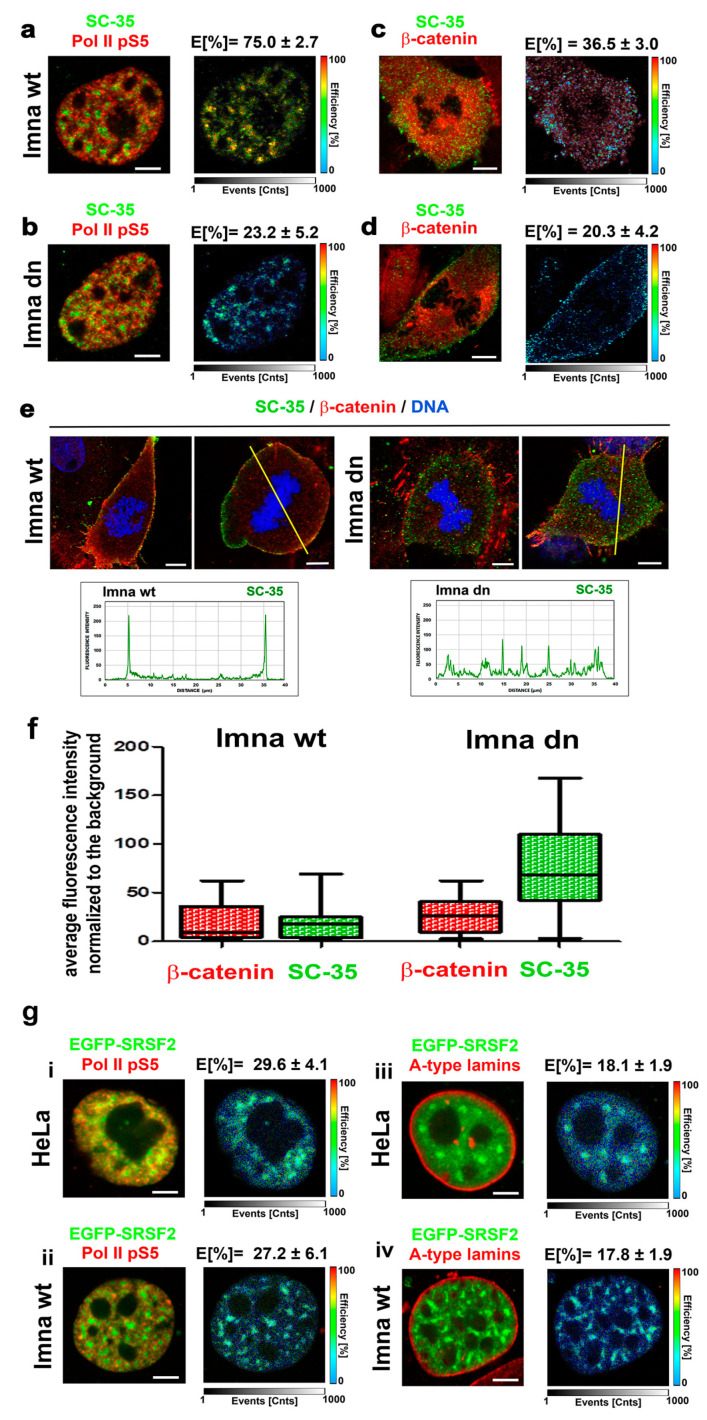
The FLIM–FRET analysis of protein-protein interactions. (**a**) In interphase lmna wt cells, the SC-35 protein interacts with the phosphorylated form of RNAP II (FRET efficiency 75.0% ± 2.7%). (**b**) The depletion of A-type lamins weakened the interaction of SC-35 and RNAP II; FRET efficiency was 23.2 ± 5.2%). (**c**) The SC-35 protein and β-catenin interacted in mitotic cells (FRET efficiency was 36.5% ± 3.0%). (**d**) The depletion of A-type lamins weakened the interaction of SC-35 and β-catenin; FRET efficiency was 20.3% ± 4.2%). (**e**) In wt MEFs, the SC-35 protein and β-catenin colocalized at the plasma membrane of mitotic cells, while depletion of the lmna gene reduced both proteins’ level (SC-35 and β-catenin), associated with the plasma membrane. See also the quantification across yellow lines, showing the reorganization of the SC-35 protein in lmna dn cells compared with the wt counterpart. (**f**) Average fluorescence intensity of the SC-35 protein and β-catenin visualized by immunohistochemistry in lmna wt and lmna dn mitotic cells; an example of cells analyzed is shown in panel e (*n* = 14 of lmna wt cells and *n* = 15 of lmna dn fibroblasts). (**g**) The degree of interaction between EGFP-tagged SRSF2 and phosphorylated form of RNAP II in (**i**) wt MEFs and (**ii**) HeLa cells. FLIM–FRET analysis for EGFP-tagged SRSF2 and lamin A in (**iii**) wt MEFs and (**iv**) HeLa cells.

**Table 1 cells-10-00297-t001:** Conditions of the cultivation of selected tumor cells and non-cancerogenic cell lines.

Cell Line	Cell Culture Medium	Reference
**A549** (human lung cancer) (#ATCC^®^ CCL-185™, Germany)	DMEM medium supplemented with 10% FCS	[30]
**HL60** (human acute promyelocytic leukemia) (#ATCC^®^ CCL-240™, Germany)	IMDM medium supplemented with 10% FCS	[32]
**MCF7** (human adenocarcinoma) (#ATCC^®^ HTB-22™, Germany)	EMEM medium supplemented with 0.01 mg/mL human recombinant insulin and 10% FCS	[33]
**MOLP8** (human multiple myeloma) (#ACC 569, DSMZ, Germany)	RPMI 1640 medium supplemented with 20% FCS	[34]
**U2OS** (human osteosarcoma) collaboration with the Institute of Biology and Medical Genetics, Charles University in Prague)	DMEM medium supplemented with 10% FCS	[35]
**U937** (human histiocytic lymphoma) (#ATCC^®^ CRL-1593.2™, Germany)	RPMI 1640 medium supplemented with 10% FCS	[34]
**HaCaT** (human keratinocytes) (#300493, CLS, Germany)	DMEM medium supplemented with 10% FCS	[36]
**IMR90** (human lung fibroblast) (#ATCC^®^ CCL-186™, Germany)	EMEM medium supplemented with 10% FCS, 1% non-essential amino acids (NEAA), and 2mM glutamine	[37]

**Table 2 cells-10-00297-t002:** Specification of donors and acceptors used for fluorescence lifetime image microscopy–Förster resonance energy transfer (FLIM-FRET) experiments.

**QY_Cy3_** **(DONOR)**	**EC_Cy5_(M^−1^cm^−1^)**	**QY_Cy5_** **(ACCEPTOR)**	**J(λ)** **(*1 × 10^15^ M^−1^cm^−1^nm^4^)**	**R_0_(Å)**	**R_0_ × QY_Cy5_**
0.15	250,000	0.30	7.6	49.73	14.92
**QY_EGFP_** **(DONOR)**	**EC _AF594_ * (M^−1^cm^−1^)**	**QY_AF594_** **(ACCEPTOR)**	**J(λ)** **(*1 × 10^15^ M^−1^cm^−1^nm^4^)**	**R_0_(Å)**	**R_0_ × QY_AF594_**
0.60	92,000	0.66	1.80	49.30	32.54

* AF594 = Alexa Fluor 594. QY = Quantum Yield; EC = Extinction Coefficient; J(λ) = Overlap Integral, $J\left(\lambda \right)=\int_{0}^{\propto }F_{D}\left(\lambda \right)\varepsilon_{A}\left(\lambda \right)\lambda^{4}d\lambda /\int_{0}^{\propto }F_{D}\left(\lambda \right)d\lambda$; R_0_ = Föster Radius, $R_{0}=0.211\sqrt[{6}]{\kappa^{2}n^{-4}Q_{D}J\left(\lambda \right)}$; n = refractive index of mounting medium (Vectashield, n = 1.45) and κ^2^ = orientation factor κ^2^ = 0.6667).

## Data Availability

Original micrographs (files in Giga bites, GB), and original western blots are on demands, please address Eva Bártová (e-mail: bartova@ibp.cz).

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
