# Peer review of "The SC-35 Splicing Factor Interacts with RNA Pol II and A-Type Lamin Depletion Weakens This Interaction"

_cells, 2021, doi:10.3390/cells10020297_

Round 1
Reviewer 1 Report
The paper by Legartoba S, et al., entitled SC-35 splicing factor interacts with RNA pol II and nuclear blebs induced by PARP inhibitor are Sc-35 positive, describes a series of immunofluorescence/confocal microscopy, including FLIM-FRET assays, and western blot experiments, with the aim of analyzing SC-35 localization and protein expression in different cell lines, including many cancer cell lines and wild type and lamin A-knock- out mouse embryonic fibroblast cultures, and under different clinically used treatments.
Major points
Across the manuscript the analysis of some proteins appears suddenly with no rationale supporting the experimental design.
This is the case for the relationship between the following proteins.
- PARP (enzyme involved in DNA repair) and the alternative splicing factor SC-35.
- Lamin A/C and SC-35.
- Beta-catenin with Sc-35
- SC35 and UBF1?
- SC-35 and 53BP1 DNA repair factor.
- SC-35 with mitotic chromosome rossettes and SMD1 and PCNA.
It is not clear to me what the rationale for this study was. The study appears to be a collection of excellent confocal images with high resolution, using available antibodies, instead of a well-designed experiments. It lacks references from the literature or argumentation to support why the authors chose the proteins and cell cultures analyzed
For instance, under what evidence the authors hypothesized that SC35 levels might be related to those of beta-catenin and lamin A; argumentation is missing.
What was the idea in mind regarding the analysis of a potential colocalization between SC35 and UBF1? Likewise, what is the meaning of the effect of lamin A deficiency on the colocalization/interaction of SC-35 and beta-catenin and RNA pol II.
In discussion the authors claimed that the goal of this study was to know the effect of some clinically used agents on cellular processes or components, such as SC-35; however it does not fit well with “Introduction” and “Abstract” It is necessary to make more uniform these section, each on to each other.
Minor points
- It is necessary to include PARP meaning in “Abstract” before using its abbreviation.
2) Lines 43, the size unit of speckles foci.it is missing
3) The magnification of Fig. 2 D is too low to see details of SC35 distribution and even the presence of nuclear blebs.
3) Fig. 2 D does not show indeed increased SC35 levels; instead it should be referred as increased immunofluorescent signal, while WB experiments do show protein levels.
4) Fig. 4 line 255. It should be say, ..treated with PARP…, not by PARP.
5) Why the WT lamin A cells exhibit nuclear blebs and micronuclei? Such structures are higher in number in lamina A deficient cells? What was the passage number of these cultures?
6) What is the reason of showing a Comassie-staining gels to observe total protein as a loading control, instead of using actin or tubulin or any other well-known loading control protein?
7) Throughout the Manuscript the authors used the sentence … treatment by or treated by. It must be say instead …treatment with.. or treated with ….See for instance lines 303 and 424.
8) Why PARP inhibitors caused nuclear blebs, hypothesis should be mentioned.
9) It is lacking a functional conclusion derived from data in “Abstract”.
What is the main question addressed by the research?
The Manuscript presents a series of immunofluorescence/confocal microscopy, including FLIM-FRET assays, and western blot experiments, with the aim of analyzing SC-35 localization and protein expression in different cell lines, including many cancer cell lines and wild type and lamin A-knock- out mouse embryonic fibroblast cultures, and under different clinically used treatments. However, it is not clear to me what the rationale for this study was. The study appears to be a collection of excellent confocal images with high resolution, using available antibodies, instead of a well-designed experiments.
Is it relevant and interesting?
The analysis of SC-35 an important splicing factor in different cell cultures and under different clinically-used agents is interesting; nevertheless, the rational of the study is not clear and then, the conclusions seem not relevant.
How original is the topic? What does it add to the subject area compared
with other published material?
The study present a number of confocal images and WB data showing SC-35 changes in distribution and protein expression as well as its colocalization with other important nuclear proteins. It is new; however, the scientific context and rationale underlying the study is missing.
Is the paper well written? Is the text clear and easy to read?
The manuscript is well-written, the problem is the lack of a rationale for the study.
Are the conclusions consistent with the evidence and arguments
presented? Do they address the main question posed?
I found description of results, instead of a conclusions. .
Recommendation
Revise. In my opinion the authors failed to communicate the purpose of the study and the rationale behind each of the designed experiments.
Major changes are required before acceptation.
Author Response
We thank the reviewers for his/her additional suggestions and interesting comments regarding how to improve our manuscript. Herein, we have replied to the reviewers’ criticisms. In the revised manuscript, changes are denoted with red fonts. Moreover, we would like to confirm that our manuscript was written according to the instructions of the Cells journal, and we acknowledge that our work has not been previously published, that it is not under consideration for publication elsewhere, and that the text has been approved by all authors.
Reviewer 1
The paper by Legartova S, et al., entitled SC-35 splicing factor interacts with RNA pol II and nuclear blebs induced by PARP inhibitor are Sc-35 positive, describes a series of immunofluorescence/confocal microscopy, including FLIM-FRET assays, and western blot experiments, with the aim of analyzing SC-35 localization and protein expression in different cell lines, including many cancer cell lines and wild type and lamin A-knock- out mouse embryonic fibroblast cultures, and under different clinically used treatments.
Major points
Across the manuscript, the analysis of some proteins appears suddenly with no rationale supporting the experimental design.
This is the case for the relationship between the following proteins.
- PARP (enzyme involved in DNA repair) and the alternative splicing factor SC-35.
- Lamin A/C and SC-35.
- Beta-catenin with SC-35
- SC35 and UBF1?
- SC-35 and 53BP1 DNA repair factor.
- SC-35 with mitotic chromosome rossettes and SMD1 and PCNA.
Answer: Primarily, a rationale of these experiments was to study the properties of the SC-35 protein in the cells exposed to DNA damage. Therefore, we studied the effect of γ-irradiation, PARP inhibitor, and A-type lamin depletion. It is well-known that inhibition of PARP increases the levels of gamma H2AX-associated DNA damage in the S phase of the cell cycle (Idun Dale Rein et al., Cell Cycle, 2015;14(20):3248-60,doi: 10.1080/15384101.2015.1085137), and similarly, in lamin A/C-deficient cell population, there is an increased frequency of cells with delayed disappearance of γ-H2AX foci and defective repair factor recruitment (Mre11, CtIP, Rad51, RPA, and FANCD2). Moreover, these authors suggested that the physiological function of A-type lamins is essential for maintaining genomic stability following replication fork stalling (Singh M et al., Mol Cell Biol. 2013 Mar; 33(6): 1210–1222 doi: 10.1128/MCB.01676-12). Thus, as the next step, we analyzed a link of SC-35 to replication factors, especially PCNA playing a role in DNA damage repair. As the marker of DNA damage, we selected the 53BP1 protein regulating non-homologous end-joining mechanism (NHEJ) (Guo X eta l., Nucleic Acids Res. 2018 Jan 25; 46(2): 689–703, doi: 10.1093/nar/gkx1208) or phosphorylation of histone H2AX (western blots). From this view, we also studied transcription factor UBF1 that was additionally found to be a factor of the Nucleotide Excision Repair (NER) (Stixová et al., Epigenetics and Chromatin, 2014).
Due to the fact that our results showed SC-35 decorating the plasma membrane of mitotic cells, we tried to find a link of this protein to some factor associated with the plasma membrane and regulating both cell-to-cell adhesion and gene expression. For that reason, we selected beta-catenin (McCrea PD, Turck CW, Gumbiner B. A homolog of the armadillo protein in Drosophila (plakoglobin) associated with E-cadherin. Science. 254 (5036): 1359–61. Bibcode: 1991Sci...254.1359M. doi:10.1126/science.1962194.
We admit that studies on the SMD1 protein are off the rationale of these experiments. Still, the selection of this protein was inspired by the observation of Dale Rein et al. (2015), showing the principles of Replication-induced DNA damage after PARP inhibition. Nevertheless, data about this protein we deleted in the revised version of the manuscript.
Inhibitor of RNAP II we studied because splicing occurs co-transcriptionally; thus, we expected changes in the distribution and functional properties of the SC-35 protein. Also, in Sorokin et al. (Nucleus, 2015), we observed a pronounced DNA damage caused by RNA pol I inhibitor, Actinomycin D. Here, this inhibitor also induced an apparent DNA damage – see the level of gH2AX in the revised Fig. 4a.
The link between nucleolar protein UBF1 and SC35 we studied not only because we recently observed UBF1 as a DNA repair-related factor, but also, we wanted to confirm that SC-35 nuclear localization is out of the compartment of nucleoli, as expected.
It is not clear to me what the rationale for this study was. The study appears to be a collection of excellent confocal images with high resolution, using available antibodies, instead of a well-designed experiments. It lacks references from the literature or argumentation to support why the authors chose the proteins and cell cultures analyzed
Answer: See rationale, explained above. HeLa tumor cells we studied due to the fact that such tumors are cured by PARPi and ionizing radiation. Lmna wt and lmna dn cells we used because we wanted to test the effect of lamin depletion on the distribution and functional properties of the SC35 protein.
For instance, under what evidence the authors hypothesized that SC35 levels might be related to those of beta-catenin and lamin A; argumentation is missing.
Answer: Regression analysis showed no correlation between the protein levels studied in distinct cell types (Fig. 5a, b); however, Fig. 5a shows that an absence of lamin A is accompanied by the highest levels of the SC-35 protein. This is the case of pluripotent hES cells (non-differentiated and partially differentiated), progenitor HL60, and U937 leukemia cells. The same trend we saw when we measured the fluorescence intensity of SC-35 in lmna dn cells, in comparison to lmna wt cells (Fig. 9f). In addition, mitotic lmna dn cells were characterized by an appearance of a dispersed form of the SC-35 protein associated with the plasma membrane, but in lmna wt cells, SC-35 formed a well-distinguished ring associated with the plasma membrane. According to these data, A-type lamins might regulate the arrangement of SC-35 domains.
In the case of beta-catenin, there is no correlation between its level and the SC-35 level in distinct cell types (Fig. 5b), but FLIM-FRET analysis showed a relatively high degree of co-localization (a high FLIM-FRET efficiency) between these proteins, studied in mitotic cells (Fig. 9c). Beta-catenin, as a potential interacting partner with SC-35, we selected due to its association with plasma membrane that was also positive on SC-35 - in mitosis.
What was the idea in mind regarding the analysis of a potential co-localization between SC35 and UBF1? Likewise, what is the meaning of the effect of lamin A deficiency on the co-localization/interaction of SC-35 and beta-catenin and RNA pol II.
Answer: The link between nucleolar protein UBF1 and SC35 we studied not only from the view that recently observed UBF1 as DNA repair-related factor (Stixova et al., 2014), but also, we want to confirm that SC-35 nuclear localization is out of the compartment of nucleoli, in which UBF1 is accumulated as the main transcription factor for ribosomal genes. It was also a verification of a well-known phenomenon.
Generally, we study a link of SC-35 to DNA repair factors, and UBF1 is one of them (Stixová e al., 2014).
We observed the SC35 positivity in nuclear blebs (Fig. 2b, c; Fig. 3) that appear in laminopathy cells. It is known that nuclear blebs are specific by their lamin A/C positivity, but lamin B negativity (LB2 negativity; Shimi et al., Genes and Development, 2008). In this case, we analyzed how A-type lamin deletion affects selected protein-protein interaction. According to these results, lamin depletion decidedly affects the level of the SC-35 protein in its interaction properties with RNA pol II. These results document that fully functional A-type lamin proteins stabilize splicing machinery that works co-transcriptionally (Fig. 5a, 9a, b, e).
However, FLIM-FRET analysis did not show a direct interaction between EGFP-SRSF2 (an alternative name SC-35) and A-type lamins.
In discussion the authors claimed that the goal of this study was to know the effect of some clinically used agents on cellular processes or components, such as SC-35; however it does not fit well with “Introduction” and “Abstract” It is necessary to make more uniform these section, each on to each other.
Answer: the discussion section was completely rewritten because, in the submitted version, we paid more attention to the function of PARP inhibitor, but the manuscript is mainly about the distribution and functional properties of the SC-35 protein.
Minor points
- It is necessary to include PARP meaning in “Abstract” before using its abbreviation.
Answer: Inhibitor of PARP [poly (ADP-ribose) polymerase] – it was inserted into the Abstract.
2) Line 43, the size unit of speckles foci.it is missing
Answer: An average size of nuclear speckles in HeLa cells is 4.79 ± 0.78 µm3 (and additionally see an example in Fig. 2a – one measurement).
3) The magnification of Fig. 2 D is too low to see details of SC35 distribution and even the presence of nuclear blebs.
Answer: The magnification of this figure was enlarged in the revised version, but in the format of Cells journal, we have to adopt the images to individual pages.
3) Fig. 2 D does not show indeed increased SC35 levels; instead it should be referred as increased immunofluorescent signal, while WB experiments do show protein levels.
Answer: It is generally accepted that the best choice for protein level quantification is western blot. To quantify immunofluorescence (IF) is also possible, but we do not trust such analysis (nevertheless, you can see our analysis in revised Fig. 2e). Western blot is more exact because it is possible to normalize the level of the given protein to the total protein level or to the level of some reference protein.
4) Fig. 4 line 255. It should be say, ..treated with PARP…, not by PARP.
Answer: This part was corrected, and the whole revised text was controlled by an advanced version of Grammarly software.
5) Why the WT lamin A cells exhibit nuclear blebs and micronuclei? Such structures are higher in number in lamina A deficient cells? What was the passage number of these cultures?
Answer: Also, in the highest (50th) passage of lmna wt cells, nuclear blebs can be visible – these cells were senescent, and it is mentioned in the revised version of the manuscript (also see Fig. 3a).
6) What is the reason for showing Coomassie-staining gels to observe total protein as a loading control, instead of using actin or tubulin or any other well-known loading control protein?
Answer: During the differentiation, cells undergo extensive genome reorganization that includes house-keeping genes/ proteins. To be able to compare/ quantify western blot from Fig. 5 (position on WB: 1-4 with the other either cancer cell lines, position 5-11 or normal cell lines - position 12-13), we optioned for total protein level instead of, e.g., actin or tubulin. We used staining of proteins by amido black on blot transfer membranes. Amido black has a sensitivity similar to that of Coomassie blue, but it stains faster. The membranes were scanned and used in the quantification via ImageJ software. For the sake of the continuity and homogeneity of the manuscript, we decided to use the total protein level in Fig. 5a and 4a. Also, according to our experience, in some cases, drugs or inhibitors affect reference proteins such as actin or tubulin. Therefore, normalization of western blot data to the total protein level is the better choice (according to our experiences).
7) Throughout the Manuscript the authors used the sentence … treatment by or treated by. It must be say instead …treatment with.. or treated with ….See for instance lines 303 and 424.
Answer: This part was corrected in the revised version.
8) Why PARP inhibitors caused nuclear blebs, hypothesis should be mentioned.
Answer: An appearance of nuclear blebs, as well as micronuclei, should be a consequence of lamin disruption and abrogation of lamin interaction with heterochromatin at the nuclear periphery. These changes in the cell nucleus can be caused by PARPi. According to some authors, cells in the S-phase of the cell cycle are more prone to nuclear budding and blebbing (Koh-ichi Utani et al., Plos One, 2011). According to Caruso et al. (Ultrasttuctura Pathology, 2010), morphological findings of these abnormal nuclear structures appear in parallel with p53 and pKi-67 overexpression caused by a faulty mitotic checkpoint or mitotic catastrophe that especially appears in cancer cells.
9) It is lacking a functional conclusion derived from data in “Abstract”.
Answer: This is our conclusion: Together, we showed a high density of the SC‑35 protein in the nuclear blebs and micronuclei, which are also sites of both transcription and splicing. This conclusion supports the fact that splicing proceeds transcriptionally, and according to our data, this process is A-type lamin dependent.
What is the main question addressed by the research?
Answer: Primarily, we studied properties of the main splicing factor, the SC-35 protein, in the cells exposed to DNA damaging agents, including clinically used PARP inhibitor and gamma-radiation. Because the physiological function of A-type lamins is essential for maintaining genome stability after DNA damage and internal lamins protect replication fork, we addressed the question of how A-type lamin depletion affects the stability of the SC-35 protein-positive nuclear speckles.
-----------------------------------------------------------------------------------------------------------------
The manuscript presents a series of immunofluorescence/confocal microscopy, including FLIM-FRET assays, and western blot experiments, with the aim of analyzing SC-35 localization and protein expression in different cell lines, including many cancer cell lines and wild type and lamin A-knock- out mouse embryonic fibroblast cultures, and under different clinically used treatments. However, it is not clear to me what the rationale for this study was. The study appears to be a collection of excellent confocal images with high resolution, using available antibodies, instead of a well-designed experiments.
Is it relevant and interesting?
The analysis of SC-35 an important splicing factor in different cell cultures and under different clinically-used agents is interesting; nevertheless, the rational of the study is not clear and then, the conclusions seem not relevant.
Answer: We studied properties of the main splicing factor, the SC-35 protein, in the cells exposed to DNA damaging agents, including clinically used PARP inhibitor. Because the physiological function of A-type lamins is essential for maintaining genome stability after DNA damage and internal lamins protect replication fork, we addressed the question of how A-type lamin depletion affects the SC-35 protein.
The extended conclusion is the following: A high density of the SC-35 protein is in nuclear blebs or micronuclei, which are reservoirs of both transcription and splicing factors. This observation fits well with the fact that SC-35 interacts with the phosphorylated form of RNAP II. In mitotic cells, there is significant interaction between SC-35 and β-catenin, and A-type lamin depletion weakens both protein-protein interactions.
How original is the topic? What does it add to the subject area compared
with other published material?
Answer: SC-35 and its structural properties were addressed and discussed in two basic papes (Misteli, T.; Spector, D.L. RNA polymerase II targets pre-mRNA splicing factors to transcription sites in vivo. Mol Cell 1999, 3, 697-705, doi:10.1016/s1097-2765(01)80002-2 and Misteli, T.; Caceres, J.F.; Spector, D.L. The dynamics of a pre-mRNA splicing factor in living cells. Nature 1997, 387, 523-527, doi:10.1038/387523a0).
Similarly, Tripathi, K.; Parnaik, V.K. Differential dynamics of splicing factor SC35 during the cell cycle. J Biosci 2008, 33, 345-354, doi:10.1007/s12038-008-0054-3. provided important data on the dynamic properties of SC-35. Since then, this protein was analyzed from the functional view, so we decided to reveal more about the nuclear distribution pattern and levels of this fundamental splicing factor.
The study present a number of confocal images and WB data showing SC-35 changes in distribution and protein expression as well as its co-localization with other important nuclear proteins. It is new; however, the scientific context and rationale underlying the study is missing.
Answer: We studied properties of the main splicing factor, the SC-35 protein, in the cells exposed to DNA damaging agents, including clinically used PARP inhibitor and gamma-radiation. Because the physiological function of A-type lamins is essential for maintaining genome stability after DNA damage, we addressed the question of how A-type lamin depletion affects distribution and the levels of the SC-35 protein.
Is the paper well written? Is the text clear and easy to read?
Answer: In the revised version, we tried to improve the style of writing, especially in the discussion section.
The manuscript is well-written, the problem is the lack of a rationale for the study.
Answer: We addressed a question of how DNA damaging agents and lamin depletion affect structural and functional properties of the splicing factor, SC-35. This rationale we mentioned in the Abstract and the last paragraph of the Introduction - see the revised version.
Are the conclusions consistent with the evidence and arguments
presented? Do they address the main question posed?
I found description of results, instead of a conclusions.
Answer: In the revised version, we explained our conclusions more appropriately.
Reviewer 2 Report
The article by Legartova, Fagherazzi and Bartova examines the changes in localization and amount of SC35 protein in response to the exposure of several cell lines to transcriptional repression. These forms of transcriptional repression include the use of PARP inhibitors, RNAP I and II inhibitors as well as g-irradiation. The authors also examine the impact of lamin A depletion on SC35 localization.
The imaging and the blotting by the authors is excellent and they demonstrate a large breadth in which there is a correlation or anti-correlation of the exposure to inhibitors. Although the imaging is excellent I am curious as to why the image sizes and magnifications are not consistent. Some of the image sizes such as those in panel D make it extremely challenging to evaluate the data.
The first line of the abstract mentions that this article examines the ‘how functional’ SC35 is. Although the localization and levels change, there is no data to imply what the functional consequence of this are or what this change in localization is implying as far as specific processes.
The title further implies this article will discuss PARP inhibitors, which it does, but also examines many more aspects. The title should be reformatted to reflect this. This title is also grammatically incorrect.
Unfortunately, I cannot recommend this article for publication at this time. With a reworking of the language to better reflect what the authors are observing and what the data does indicate (and what it does not indicate). I would be happy to see it again; however, once the issues raised above and below are addressed. The lack of the significance of these changes is the major issue for me. This is excellent phenomenon but why this is important and what it means will lead to a better impression.
Major comments:
As mentioned above, this is very much a localization paper. Some great images but the 'function' was not examined in any of the data. Certainly a change in localization but what is the significance of this? Likely a change in transcriptional status. What is the change in transcriptional (global) in response to these treatments? Why is the levels of the A type lamins linked to SC-35 levels?
The word significant is used throughout the manuscript; however, I can find no evidence of statistical analysis. This maybe personal preference but if something has not been shown statistically, it can not be called significant.
The last line of the abstract - again what is the functional significance of this? There are some great images here and the blots look great but I am not sure what the importance of this is. I think some cleaning up of the language and clear statements of why this of interest would significantly help this paper.
Figure 2: Why are all the images different sizes and magnifications? In particular panel D. This makes the data extremely hard to interpret or draw conclusions from.
Figure 4: T test to compare against untreated controls or an ANOVA test to indicate if these were different from one another? Also the numbers are not clear at the bottom? What do these represent?
Line 280: Is this the active form or is this a storage site for excess currently un-utilized protein? If un-utilized then what would be the functional significance of this co-localization?
Line 281: So they were not connected and therefore do not co-localize? I am not seeing the significance of this here.....
Figure 8: Scale bar for inset images. This does not look like co-localization to me. this looks like one domain is beside the other. Also are these cells synchronous? Are they S-phase or G1/G2 cells? A replication factor would certainly change localization patterns in response to this.
Minor comments:
DAPI does not just bind AT rich sequences; it binds these bases easier but not exclusively. I assume the authors want to indicate that the DAPI staining pattern is due to decreased chromatin content within these regions.
Line 241: For statements like this the authors could add phrases such as 'By both IF and western blotting, we demonstrate that SC-35 protein .....' I was about to write a comment about westerns and then saw the figure. This type of phrasing will help with removing doubts from the readers mind.
Line 278: ‘It was the truth…’ this should be changed to something like ‘It was observed…’ Truth has a different philosophical meaning that is not applicable here.
There are grammatical errors throughout that need to be addressed.
Author Response
Reviewer 2
The article by Legartova, Fagherazzi and Bartova examines the changes in localization and amount of SC35 protein in response to the exposure of several cell lines to transcriptional repression. These forms of transcriptional repression include the use of PARP inhibitors, RNAP I and II inhibitors as well as g-irradiation. The authors also examine the impact of lamin A depletion on SC35 localization.
The imaging and the blotting by the authors is excellent and they demonstrate a large breadth in which there is a correlation or anti-correlation of the exposure to inhibitors. Although the imaging is excellent I am curious as to why the image sizes and magnifications are not consistent. Some of the image sizes such as those in panel D make it extremely challenging to evaluate the data.
Answer: When we wanted to show a detail of the cell nucleus, we used a larger image, but in the case of the whole cell population, we show several cell nuclei (see Fig. 2d). In the original version, we did not quantify fluorescence intensity in Fig. 2d (but see revised Fig. 2e) because we think that western blot provides better quantification of protein levels (see Fig. 4a, b).
The first line of the abstract mentions that this article examines the ‘how functional’ SC35 is. Although the localization and levels change, there is no data to imply what the functional consequence of this are or what this change in localization is implying as far as specific processes.
Answer: It is the truth that we instead analyzed nuclear distribution profiles and the level of the SC-35 protein, but we think that changes in the protein level have functional consequences.
The title further implies this article will discuss PARP inhibitors, which it does, but also examines many more aspects. The title should be reformatted to reflect this. This title is also grammatically incorrect.
Answer: The title was rewritten in the following way: The SC-35 splicing factor interacts with RNA pol II and A-type lamin depletion weakens this interaction. For language correction, we used the Grammarly software that did not show grammatical errors, but the revised version is corrected by the linguistic service, the American Journal Experts (AJE).
Unfortunately, I cannot recommend this article for publication at this time. With a reworking of the language to better reflect what the authors are observing and what the data does indicate (and what it does not indicate). I would be happy to see it again; however, once the issues raised above and below are addressed. The lack of the significance of these changes is the major issue for me. This is excellent phenomenon but why this is important and what it means will lead to a better impression.
Answer: Manuscript was rewritten, and some additional experiments were performed. Grammar was corrected by the linguistic service, the American Journal Experts (AJE), and Grammarly software.
Major comments:
As mentioned above, this is very much a localization paper. Some great images but the 'function' was not examined in any of the data. Certainly a change in localization but what is the significance of this? Likely a change in transcriptional status. What is the change in transcriptional (global) in response to these treatments? Why is the levels of the A type lamins linked to SC-35 levels?
Answer: We showed that lamin depletion affects the level of the SC-35 protein (Fig. 5a) and that lamin depletion weakens SC-35 interaction with RNA pol II (Fig. 9a, b). These results imply that disorder in the function of A-type lamin proteins destabilizes spicing factors, including SC-35. The effect of A-type lamins on SC‑35 level and distribution is likely indirect because FLIM-FRET showed a relatively low probability of interaction between SC-35 (EGFP-SRSF2) and lamin A (Fig. 9g).
The word significant is used throughout the manuscript; however, I can find no evidence of statistical analysis. This maybe personal preference but if something has not been shown statistically, it cannot be called significant.
Answer: Word significance was re-evaluated, and statistical tests like regression analysis were verified. For statistical analysis, we used the Mann-Whitney U-test.
The last line of the abstract - again what is the functional significance of this? There are some great images here and the blots look great but I am not sure what the importance of this is. I think some cleaning up of the language and clear statements of why this of interest would significantly help this paper.
Answer: Abstract was rewritten; the main result is mentioned in the title.
Together, we showed a high density of the SC-35 protein in the nuclear blebs or micronuclei, which are sites of both transcription and splicing factors. This observation fits well because SC-35 interacts with the phosphorylated form of RNAP II; moreover, this interaction is A-type lamin dependent.
Figure 2: Why are all the images different sizes and magnifications? In particular panel D. This makes the data extremely hard to interpret or draw conclusions from.
Answer: Different sizes we used when we wanted to show a detail of the cell nucleus, we used a larger image, but in the case of the whole cell population, we show several cell nuclei, which looks like low resolution (see Fig. 2b-d). In the first version, we did not quantify fluorescence intensity in Fig. 2d. Still, for revision, we performed quantification that shows an increased level of the SC‑35 protein after alpha-amanitin treatment (see revised Fig. 2e). However, we think that western blot provides better quantification of the protein levels (see Fig. 4a, b).
Figure 4: T test to compare against untreated controls or an ANOVA test to indicate if these were different from one another? Also the numbers are not clear at the bottom? What do these represent?
Answer: The numbers in Fig. 4a, b were replaced by the description of treatments, and instead ANOVA test, we performed the Mann-Whitney U-test.
The Mann-Whitney U test (STATISTICA software) is a nonparametric test of the null hypothesis that is applied for X and Y values, randomly selected from two experimental units. The primary step of this analysis contains the U statistic involving the use of asymmetric real-valued function h(x,y). The following formula describes the approach:
- (n 2) is the binomial coefficient,
- (ut) are independent and identically distributed variables,
- Σ = summation notation
The following two formulas are applicable for the Mann-Whitney U Test. R is the sum of ranks in the sample, and n is the number of items in the sample.
We ran the test at the 5% level of significance (i.e., * means α=0.05).
Line 280: Is this the active form or is this a storage site for excess currently un-utilized protein? If un-utilized then what would be the functional significance of this co-localization?
Answer: This part was corrected in the revised version. We are aware that protein co-localization does not always mean mutual functional interaction. Thus, we additionally perform FLIM-FRET analysis. We believe that when there is a high FLIM-FRET efficiency (E≥25%), there is a high probability that studied proteins interact and work in the same regulatory pathway, and they are functionally compatible.
Line 281: So they were not connected and therefore do not co-localize? I am not seeing the significance of this here.....
Answer: This part was corrected in the revised version. Foci are in close proximity, but there was no co-localization.
Figure 8: Scale bar for inset images. This does not look like co-localization to me. this looks like one domain is beside the other. Also are these cells synchronous? Are they S-phase or G1/G2 cells? A replication factor would certainly change localization patterns in response to this.
Answer: The information about the SMD1 protein we deleted in the revised version. We focused on DNA repair proteins, so we only analyzed PCNA playing a role in both DNA replication and DNA repair. According to the nuclear distribution profile of PCNA, we show and analyzed cells in the non-S phase and in the late S-phase (see revised Fig. 8c). In these cases, we studied the co-localization rate between PCNA and SC-35 proteins (Fig. 8d). Leica software co-localization tool showed a relatively high value calculated for Pearson’s correlation coefficient, especially in the cells analyzed in the S-phase of the cell cycle (Fig. 8d).
From the view of cell synchronization, we have experiences that a pronounced DNA damage appears. Thus, when we perform DNA repair studies (tests of DNA damaging agents), we do not apply cell synchronization procedure, like relatively gentle serum deprivation, because we do not want to potentiate additional DNA damage. Note: Cell cycle phases we are able to distinguish according to PCNA profiles or by the use Fucci cellular system (as shown in our previous papers).
Minor comments:
DAPI does not just bind AT rich sequences; it binds these bases easier but not exclusively. I assume the authors want to indicate that the DAPI staining pattern is due to decreased chromatin content within these regions.
Answer: We agree entirely with the reviewer, who pointed out that a low DAPI density is due to decreased chromatin content within these genomic regions. This part was corrected in the revised version.
Line 241: For statements like this the authors could add phrases such as 'By both IF and western blotting, we demonstrate that SC-35 protein .....' I was about to write a comment about westerns and then saw the figure. This type of phrasing will help with removing doubts from the readers mind.
Answer: Many thanks to the reviewer. In the revised version, we verify this claim, as recommended. The whole manuscript was carefully revised and rewritten.
Line 278: ‘It was the truth…’ this should be changed to something like ‘It was observed…’ Truth has a different philosophical meaning that is not applicable here.
Answer: This part was corrected.
There are grammatical errors throughout that need to be addressed.
Answer: The revised manuscript was corrected by the American Journal Experts – a linguistic service and by Grammarly software.
Reviewer 3 Report
This is a very comprehensive study studying the effects of a PARP inhibitor, gamma-irradiation, Pol1 and Pol2 inhibitors on SC-35 localization and expression using a multitude of human cell lines. They’ve used ab11826 as their SC-35 antibody, which is the original SC-35 antibody described by Fu & Maniatis.
Their imaging data show enhanced staining of SC-35 in PARPi treated cells, and a reduction in SC-35 staining treated with gamma-irradiation. The image with alpha-amanitin treated cells is really small to be able to correctly interpret, but it looks like the SC-35 signal becomes rather diffuse.
A-type lamin depletion seems to also affect SC-35 staining, the signal looks less “specklish” and more diffuse.
Authors try to correlate the imaging data with immunoblots by looking at the 35kDa signal revealed by SC-35. The most convincing part of this blot (Fig. 4a) is the reduction in the levels of the 35 kDa signal in gamma-irradiated cells, which correlates well with their IF data (Fig. 2b).
Authors also investigate this 35 kDa band in various cell lines, and find out that hESCs, differentiated or undifferentiated have the highest expression, and some anti-correlation with lamin A signal.
The study is overall well executed, the authors thoroughly characterize SC-35 signal using immunofluorescence microscopy and western blotting (35 kDa). PARP inhibitors are a group of clinically relevant chemicals, and therefore a thorough characterization of SC-35 signal in cells treated with PARPi can be relevant to clinicians.
Major comments:
There is one major issue in the manuscript, which fundamentally affects the conclusions of the paper. This is not really the fault of the authors, but they’re affected by it regardless.
The SC-35 antibody, as it has been shown recently by Ilik et al. (doi: 10.7554/eLife.60579), seems to have been severely mischaracterized, and unlike the original publications describing it (Fu & Maniatis 1990, Fu & Maniatis 1992), and subsequent publications, it seems to have little, if anything, to do with SRSF2. For details, please take a look at the publication itself, but in summary, it looks like the primary target of the SC-35 antibody is a large, highly disordered protein called SRRM2, which, unlike SRSF2, is an integral part of the Spliceosome which has been used by Fu & Maniatis to immunize mice to raise this antibody (Fu and Maniatis, 1990). Therefore, all immunofluorescence signal presented in this manuscript using SC-35 mAb is likely originating from SRRM2 and not SRSF2. Furthermore, Ilik et al. also shows that the 35 kDa band detected by SC-35 mAb on immunoblots is almost entirely originating from SRSF7, and certainly not SRSF2. It is possible to detect SRRM2 by the SC-35 mAb in immunoblots, but to do so authors have to make sure that they are able transfer high-molecular weight proteins to the membranes they use.
As it stands, this manuscript would add up to the large amount of papers that inaccurately claim the role/distribution/expression of SRSF2 upon various stimuli/treatment using the SC-35 mAb. It may be infeasible to repeat all of the experiments in this manuscript using antibodies that are actually specific to SRSF2, SRSF7 and SRRM2. However, for the sake of their own research, I would highly recommend at least repeating some of the key experiments using SRRM2 antibodies to see if they will have the same observations.
In the absence of such experiments, I highly recommend discussing the implications of Ilik et al. on the interpretation of their results. Ideally, this should be reflected also early on in the manuscript, such that the reader understands that they are likely looking at SRRM2 in the immunofluorescence data, and SRSF7 in the immunoblotting data. In this way, researchers that might want to replicate these results, and possibly build on them will have the right tools to differentiate between SRSF2, SRSF7 and SRRM2 going forward.
Minor comment:
First sentence of the abstract: “An essential component of the splicing machinery is the SC-35 protein”, is not accurate even if it’s referring to SRSF2. The very original paper that claims this (Fu & Maniatis, 1990) uses SC35 mAb to block splicing in an in vitro splicing reaction, relevance of which hasn’t been explored in vivo since then.
Author Response
Reviewer 3:
This is a very comprehensive study studying the effects of a PARP inhibitor, gamma-irradiation, Pol1 and Pol2 inhibitors on SC-35 localization and expression using a multitude of human cell lines. They’ve used ab11826 as their SC-35 antibody, which is the original SC-35 antibody described by Fu & Maniatis.
Their imaging data show enhanced staining of SC-35 in PARPi treated cells, and a reduction in SC-35 staining treated with gamma-irradiation. The image with alpha-amanitin treated cells is really small to be able to correctly interpret, but it looks like the SC-35 signal becomes rather diffuse.
Answer: Fig. 2c showing alpha amanitin staining was enlarged, as possible. The intensity of fluorescently stained SC-35 protein was quantified in non-treated and alpha-amanitin-treated cells (Fig. 2e) and compared with western blot data in revised Fig. 4b.
A-type lamin depletion seems to also affect SC-35 staining, the signal looks less “specklish” and more diffuse.
Answer: It is the truth that in A-type lamin depleted cells, SC-35 positive regions were more diffuse in comparison to SC_35-positive speckles in lmna wt cells. Many thanks, reviewer, for this note that we mentioned in the revised version.
Authors try to correlate the imaging data with immunoblots by looking at the 35kDa signal revealed by SC-35. The most convincing part of this blot (Fig. 4a) is the reduction in the levels of the 35 kDa signal in gamma-irradiated cells, which correlates well with their IF data (Fig. 2b).
Answer: We quantified additional western blots, and in the case of the SC-35 protein, the Man Whitney U-test, showed a significant increase in the cells treated with PARPi and alpha-amanitin or Act-D, but a protein decrease after gamma-irradiation was not, unfortunately, significant (see revised Fig. 4b).
Authors also investigate this 35 kDa band in various cell lines, and find out that hESCs, differentiated or undifferentiated have the highest expression, and some anti-correlation with lamin A signal.
Answer: In this set of experiments, it works. For explanation: In partially differentiated hESCs (retinoic acid addition for 48 h), only a slight increase of lamin A/C appeared (in comparison to non-differentiated counterpart). This is a sign of differentiation, but in both non-differentiated and differentiated hESCs, there was a very low level of A-type lamins in comparison to other (more differentiated) cell types (except leukemia cell lines). Importantly, leukemia HL60 and U937 progenitor cells (non-differentiated) were characterized by very low levels of A-type lamins, which are identical with pluripotent hESCs (Fig. 5a). In these cells, we observed the highest level of the SC-35 protein in comparison to other cell types (HT29, A549, HeLa, MCF7, MOLP8, HaCaT, IMR90).
Also, we observed that lamin depletion weakens SC-35 interaction with RNAPII, as shown by FLIM-FRET, so again these results imply that changes in A-type lamins, to some extent, have an impact on the SC35 protein levels and nuclear distribution.
The study is overall well executed, the authors thoroughly characterize SC-35 signal using immunofluorescence microscopy and western blotting (35 kDa). PARP inhibitors are a group of clinically relevant chemicals, and therefore a thorough characterization of SC-35 signal in cells treated with PARPi can be relevant to clinicians.
Answer: After PAPRi, we see an appearance of the SC-35 positive nuclear blebs and an increase in the level of the SC-35 protein. Nothing else. These phenomena were observed in HeLa cervical adenocarcinoma cells; thus, the selected cell type is relevant to the target tissue, clinically treated by PARPi. We believe that such information should be valuable for clinical approaches – a combination of PARPi with gamma-radiation maintains the SC-35 protein level comparable with non-treated cells.
Major comments:
There is one major issue in the manuscript, which fundamentally affects the conclusions of the paper. This is not really the fault of the authors, but they’re affected by it regardless.
The SC-35 antibody, as it has been shown recently by Ilik et al. (doi: 10.7554/eLife.60579), seems to have been severely mischaracterized, and unlike the original publications describing it (Fu & Maniatis 1990, Fu & Maniatis 1992), and subsequent publications, it seems to have little, if anything, to do with SRSF2. For details, please take a look at the publication itself, but in summary, it looks like the primary target of the SC-35 antibody is a large, highly disordered protein called SRRM2, which, unlike SRSF2, is an integral part of the Spliceosome which has been used by Fu & Maniatis to immunize mice to raise this antibody (Fu and Maniatis, 1990). Therefore, all immunofluorescence signal presented in this manuscript using SC-35 mAb is likely originating from SRRM2 and not SRSF2. Furthermore, Ilik et al. also shows that the 35 kDa band detected by SC-35 mAb on immunoblots is almost entirely originating from SRSF7, and certainly not SRSF2. It is possible to detect SRRM2 by the SC-35 mAb in immunoblots, but to do so authors have to make sure that they are able transfer high-molecular weight proteins to the membranes they use.
As it stands, this manuscript would add up to the large amount of papers that inaccurately claim the role/distribution/expression of SRSF2 upon various stimuli/treatment using the SC-35 mAb. It may be infeasible to repeat all of the experiments in this manuscript using antibodies that are actually specific to SRSF2, SRSF7 and SRRM2. However, for the sake of their own research, I would highly recommend at least repeating some of the key experiments using SRRM2 antibodies to see if they will have the same observations.
Answer: In the revised manuscript, we performed experiments with antibodies specific to SRRM2 and SRSF7 (SRRM2 should be recognized by Ab against SC‑35 as published and discussed by Ilik et eLife, 2020). Moreover, we analyzed the kinetics of EGFP-tagged SRSF2 by FRAP methodology (see revised Fig. 4c). In this case, 24-hour of alpha-amanitin treatment potentiates the recovery time after photobleaching of EGFP-tagged SRSF2. Western blots showed that an Act-D increased the level of SRSF7, but the levels of SRRM2 were unchanged in all cases tested (see revised Fig. 4a, b). These results were inserted into the revised version of the manuscript.
In the revised version (Discussion), we mentioned that Ilik et al., eLife (2020) showing that monoclonal antibody against nuclear speckles, used here, preferentially recognizes the SRRM2 protein, but not SRSF2, as it was published in the past (Fu and Maniatis, 1992). Ilik et al. (2020) additionally documented that especially the SON factor, together with SRRM2, are responsible for nuclear speckles formation.
In mitotic cells, anti-SRRM2 and anti-SRSF7 did not reveal these proteins in MIGs. Interestingly, in telophase and early cytokinesis, SRRM2 formed robust foci in the cytoplasm. As expected, in the interphase, SRRM2 was accumulated in nuclear speckles and micronuclei. SRSF7 was homogeneously dispersed through the nucleoplasm of the interphase cells (Fig. 8e). Together, antibodies tested (anti-SC35, anti-SRRM2, and anti-SRSF7) show a distinct distribution pattern, especially in mitotic cells. In interphase, anti-SC-35 and anti-SRRM2 showed identical speckle-like structures (Fig. 1b, 8e).
In the absence of such experiments, I highly recommend discussing the implications of Ilik et al. on the interpretation of their results. Ideally, this should be reflected also early on in the manuscript, such that the reader understands that they are likely looking at SRRM2 in the immunofluorescence data, and SRSF7 in the immunoblotting data. In this way, researchers that might want to replicate these results, and possibly build on them will have the right tools to differentiate between SRSF2, SRSF7 and SRRM2 going forward.
Answer: In the revised version, we performed additional western blots with an antibody against SRRM2 and SRSF7. Also, we performed FRAP analysis showing that after alpha-amanitin treatment (24 h after the treatment), the recovery time after photobleaching of GFP-SRSF2 is more rapid than non-treated cells. Other results are described more appropriately in the paragraph above. As recommended, data of Ilik et al. we discuss in the revised version.
Minor comment:
First sentence of the abstract: “An essential component of the splicing machinery is the SC-35 protein”, is not accurate even if it’s referring to SRSF2. The very original paper that claims this (Fu & Maniatis, 1990) uses SC35 mAb to block splicing in an in vitro splicing reaction, relevance of which hasn’t been explored in vivo since then.
Answer: This sentence was revised according to the reviewer’s suggestion, as mentioned above.
Round 2
Reviewer 1 Report
The revised version of the Manuscript was substantially improved and all my observations were took into account. The study now sounds logical between strategy and results, and the Discussion section was enriched consequently. In my opinion, it deserves publishing in Cells.
Author Response
Reviewer 1
The revised version of the Manuscript was substantially improved and all my observations were took into account. The study now sounds logical between strategy and results, and the Discussion section was enriched consequently. In my opinion, it deserves publishing in Cells.
Answer: We thank the reviewers for his/her suggestions for improving our paper, and many thanks for the final recommendation.
Reviewer 2 Report
The revised manuscript by Legartova and colleagues has addressed many of the concerns from the original draft.
Major comments:
There is a lack of clear hypothesis in the manuscript. What was to first prediction or question to be addressed? A clear statement of this will help focus the readers attention to the main points.
For the quantification of the western blots, it is indicated that this was measured against total protein with images presented of sections of gel/blots; however, there is no methodology of how this was done. Was it all the bands on the gel/blot or just a specific region (if a specific region then the molecular weight marks should be shown)? How was total protein detected (antibody or other method)?
In Figure 2E it is indicated that total cell fluorescence intensity was measured. However, in the images the signal for several proteins is saturated. I am sure the author's would agree that if the intensity of the images shown were measured this would not be a true representation of the data as this would not give the true range. I request that an image representing non-saturated levels be incorporated to give the reader a true impression of what this distribution looks like and the levels that are present.
Author Response
Reviewer 2
We thank the reviewers for his/her additional suggestions and comments regarding how to improve our Manuscript. In the 2nd revision, we have replied to the reviewer's comments, and changes are denoted with green fonts.
The revised Manuscript by Legartova and colleagues has addressed many of the concerns from the original draft.
Major comments:
There is a lack of clear hypothesis in the Manuscript. What was to first prediction or question to be addressed? A clear statement of this will help focus the readers attention to the main points.
Answer: We expected that DNA damaging agents changed the nuclear architecture that is maintained by lamin proteins. Thus, we addressed how distribution properties of nuclear speckles are affected by DNA damaging agents and A-type lamin deficiency. This statement was better specified in the Abstract and the last paragraph of the Introduction section.
In detail: Primarily, a rationale of these experiments was to study the properties of the SC-35 protein in the cells exposed to DNA damage. Therefore, we analyzed the effect of γ-irradiation, PARP inhibitor, and A-type lamin depletion. It is well-known that inhibition of PARP increases the levels of gamma H2AX-associated DNA damage in the S phase of the cell cycle (Idun Dale Rein et al., Cell Cycle, 2015;14(20):3248-60, doi: 10.1080/15384101.2015.1085137), and similarly, in lamin A/C-deficient cell population, there is an increased frequency of cells with delayed disappearance of γ-H2AX foci and defective recruitment of repair factors, including Mre11, CtIP, Rad51, RPA, and FANCD2.
For the quantification of the western blots, it is indicated that this was measured against total protein with images presented of sections of gel/blots; however, there is no methodology of how this was done. Was it all the bands on the gel/blot or just a specific region (if a specific region then the molecular weight marks should be shown)? How was total protein detected (antibody or other method)?
Answer: The western blot membranes were stained with amino acid staining azo dye (amido black 10B) to stain for total protein level on transferred membrane blots. The advantage of total protein stains is that they are less sensitive than antibody-based immunodetection and therefore less likely to result in the oversaturated signal. Also, total protein stains resolve the variation in housekeeping proteins' expression that can change due to developmental changes, post-transcriptional regulation, or cell-type differences.
The normalization to total protein level considers all of the sample proteins, are visualized, and their total abundance serves for normalization. That is why no specific regions are needed, and also, no molecular weight marks need to be shown.
The quantification itself was performed using ImageJ software as following; first, we selected the area (rectangular, identical size) around the bands. ImageJ software produced histograms (one histogram per band) with numerical values that were further analyzed.
In Figure 2E it is indicated that total cell fluorescence intensity was measured. However, in the images the signal for several proteins is saturated. I am sure the author's would agree that if the intensity of the images shown were measured this would not be a true representation of the data as this would not give the true range. I request that an image representing non-saturated levels be incorporated to give the reader a true impression of what this distribution looks like and the levels that are present.
Answer: Quantification was done on non-over saturated images, just only for image presentation and printing, we increased the saturation in whole Fig. 2 to make nuclear speckles more visible after printing.
An increased level of the SC-35 protein after alpha amanitin treatment is visible not only in immunofluorescence data (Fig. 2d, e) but also we observed the same results by western blots (Fig. 4b).
Reviewer 3 Report
The manuscript has definitely improved after revision, however there are still few corrections to be made. Here are the comments and suggestions:
- There is this following sentence in the abstract which needs to be corrected:
“Antibody against the SRRM2 protein that should be recognized by anti-SC-35 (Ilik et al. 2020; [1]) showed that nuclear speckles are formed in the cytoplasm the late telophase and at the stage of early cytokinesis.”
It is not the SRRM2 antibody that is recognized by anti-SC-35, but it is anti-SC-35 that recognizes SRRM2 protein.
- In Figure 4 panel b, SRRM2 is labeled as 34kDa protein. The major isoform of SRRM2 is theoretically a 300kDa protein which migrates heavier at 330kDa due to multiple phosphorylation sites. According to Uniprot database, there is an isoform3 (identifier: Q9UQ35-3) with theoretical 34kDa weight but it has not been addressed previously if the epitope of the anti-SC-35 can recognize this isoform or not. Therefore, in this context the SRRM2 blot provides no extra information with regards to SC35 staining in IF experiments or SC-35 western blots. Especially, when interpreting the results written as: “actinomycin D (ActD), also enhanced the SC-35 protein level (Fig. 4b). However, the antibody against SRRM2 did not show the changes.” it becomes clear that SC-35 blot 35kDa band is following the trend of SRSF7 (as also reported by Ilik et al.) but not SRRM2. So, in that sense the increase that the authors see is not SC-35 (or SRRM2) but SRSF7. Which actually further validates the findings of Ilik et al.
- It seems a bit unexpected that the authors did not detect SRRM2 in MIGs of the mitotic cells, as it is shown for SC-35. Since, SC-35 is an SRRM2 antibody, it is expected that they mirror each other. Moreover, Rai et al. (Nature volume 559, pages211–216(2018)) used SRRM2-mCherry constructs interchangeable with SC-35 in their analysis. Legartová et al. do not show the similar captions for SC35 and SRRM2 staining in Figure 8, thus makes it harder to conclude which factor is in MIGs.
Author Response
Reviewer 3
We thank the reviewers for his/her additional suggestions and comments regarding how to improve our Manuscript. In the 2nd revision, we have replied to the reviewer's comments, and changes are denoted with green fonts.
The Manuscript has definitely improved after revision, however there are still few corrections to be made. Here are the comments and suggestions:
- There is this following sentence in the Abstract which needs to be corrected:
"Antibody against the SRRM2 protein that should be recognized by anti-SC-35 (Ilik et al. 2020; [1]) showed that nuclear speckles are formed in the cytoplasm the late telophase and at the stage of early cytokinesis."
It is not the SRRM2 antibody that is recognized by anti-SC-35, but it is anti-SC-35 that recognizes SRRM2 protein.
Answer: This part in the Abstract was corrected in the revised version; the abstract was rewritten. In the rest of the text, this claim seems to be OK.
We stated that according to (Ilik et al. 2020; [1]) anti-SC-35 recognizes SRRM2 protein.
By anti-SRRM2 (#HPA041411, Merck, Germany) and anti-SC-35 (#ab11826, Abcam, UK), we showed an identical morphology of nuclear speckles (in interphase) and location of nuclear speckles in nuclear blebs, but only anti-SRRM2 showed that speckles are already established in the cytoplasm of the late telophase and at the stage of early cytokinesis (Fig. 1b, 2a, 8e). This is also in accord with the observation of Galganski et al. (2017), showing nuclear speckles disassemble during the early stages of mitosis and assemble back in telophase (see discussion section). Similarly, Rai et al. (2018) (as the reviewer recommended) showed SRRM2–mCherry granule formation in mitotic cells, treated with GSK-626616 compound, an inhibitor of the dual-specificity kinase DYRK3.
Here, we would like to declare that in many papers, scientists use anti-SC-35, and here, we tried to verify our data by the additional tool, like anti-SRRM2 (#HPA041411, Merck, Germany and #SAB2108778, Merck, Germany), and EGFP-tagged-SRSF2. What is the sure, from our results, that both anti-SC35 and anti-SRRM2 determine nuclear speckles in interphase and their location in nuclear blebs, as mentioned above.
The obstacle in the use of antibodies is the fact that distinct protein epitope can be recognized, so distinct antibodies against the identical protein could reveal a distinct protein characteristic. We have to admit that all molecular-biology methods have limitations.
- In Figure 4 panel b, SRRM2 is labeled as 34kDa protein. The major isoform of SRRM2 is theoretically a 300 kDa protein which migrates heavier at 330 kDa due to multiple phosphorylation sites. According to Uniprot database, there is an isoform 3 (identifier: Q9UQ35-3) with theoretical 34kDa weight but it has not been addressed previously if the epitope of the anti-SC-35 can recognize this isoform or not. Therefore, in this context the SRRM2 blot provides no extra information with regards to SC35 staining in IF experiments or SC-35 western blots. Especially, when interpreting the results written as: "actinomycin D (ActD), also enhanced the SC-35 protein level (Fig. 4b). However, the antibody against SRRM2 did not show the changes." it becomes clear that SC-35 blot 35kDa band is following the trend of SRSF7 (as also reported by Ilik et al.) but not SRRM2. So, in that sense the increase that the authors see is not SC-35 (or SRRM2) but SRSF7. Which actually further validates the findings of Ilik et al.
Answer: If (according to Ilik et al., 2020) anti-SC35 recognizes SRRM2, the detected size of SRRM2 should be 35-34 kDa (which is the size of SC-35; 35 kDa).
It seems to be the truth, as stated by the reviewer, that by western blots, we did not see SC-35 (or SRRM2) but SRSF7. Which actually further validates the findings of Ilik et al. This claim we inserted into the revised version of the Manuscript.
From the view of antibodies use, we would like to clarify:
We purchased antibodies from Merck (Germany) anti-SRRM2 (#SAB2108778, Merck, Germany, suitable for western blot and producer declares band of 34 kDa) and it recognizes C-terminal region of Human SRRM2 (isoform 3) DKKEKSATRP SPSPERSSTG PEPPAPTPLL AERHGGSPQP LATTPLSQEP. Another anti-SRRM2 (#HPA041411, Merck, Germany, suitable for immunofluorescence) recognizes region RSRSSSPVTE LASRSPIRQD RGEFSASPML KSGMSPEQSR FQSDSSSYPT VDSNSLLGQS RLETAESKEK MALPPQEDAT ASPPRQKDKF SPFPVQDRPE SSLVFK.
The epitope is strongly dependent on producers, for example, for anti-SRRM2:
- https://www.abcam.com/srrm2-antibody-ab122719.html, corresponding to amino acids 1098 - 1203 of Human SRRM2 Isoforms 1 and 2 (Q9UQ35) and is missing in Isoform 3. Suitable for immunofluorescence analysis, b on the other hand, not suitable for western blot analysis.
- https://www.thermofisher.com/antibody/product/SRRM2-Antibody-Polyclonal/PA5-68009. Suitable only for western blot analysis (~300 kDa), but no epitope included.
The reviewer is completely right that SRRM2 (Ser/Arg-Related Nuclear Matrix Protein) is 300 kDa protein as shown at https://www.genecards.org/cgi-bin/carddisp.pl?gene=SRRM2. So, here we can only say that by anti-SRRM2 (#SAB2108778, Merck, Germany, we studied the C-terminus domain DKKEKSATRPSPSPERSSTGPEPPAPTPLLAERHGGSP QPLATTPLSQEP epitope.
It seems a bit unexpected that the authors did not detect SRRM2 in MIGs of the mitotic cells, as it is shown for SC-35. Since, SC-35 is an SRRM2 antibody, it is expected that they mirror each other. Moreover, Rai et al. (Nature volume 559, pages211–216(2018) used SRRM2-mCherry constructs interchangeable with SC-35 in their analysis. Legartová et al. do not show the similar captions for SC35 and SRRM2 staining in Figure 8, thus makes it harder to conclude which factor is in MIGs.
Answer: For our analysis, we used antibodies that were also used by many other authors (de Chiara C et al., 2009; Blanco AM et al., 2009; Park R et al., 2014; Underwood et al., 2016; Jain A & Vale RD, 2017; Zanini et al., 2017; Francis et al., 2020; etc.). SRRM2-mCherry should show a different protein distribution profile than endogenous proteins. In our paper, we analyze endogenous proteins. There is s rule with experiments with exogenous proteins that data must be verified on an endogenous level. For example, when you compare our results on EGFP-SRSF2 (Fig. 4C) and endogenous SC-35 protein in Fig. 2A, you can see that EGFP-visualized nuclear speckles are not so pronounced in comparison to endogenous once. The morphology and distribution pattern can be cell-type specific (see enclosed figure). For the revision, we also studied EFGP-SRSF2 in mitotic cells, and we observed a high density of exogenous SRSF2 in the cytoplasm of mitotic cells; moreover, the highest density of GFP-SRSF2 was between metaphase rosettes (see revised Fig. 8g).
Here, we would like to declare that in many papers, scientists use anti-SC-35, and here, we try to very our data with additional tools, like anti-SRRM2 and EGFP-tagged-SRSF2. It is identical that both anti-SC35 and anti-SRRM2 recognized nuclear speckles in interphase and their location in nuclear blebs. In mitotic cells, we have a different result when using anti-SC-35 and anti-SRRM2. Experiments were performed by Sona Legartova, working on and optimizing immunodetection for more than 15 years, so as corresponding author EB, I trust these results. It can be a question with the protein epitope we study by the use of a given antibody.
Rai et al. (2018), in supplementary Extended Data Fig. 3a showed a high density of SC-35 in the cytoplasm of mitotic cells, which agrees with our data. Also, the periphery of these cells is very dense on the SC-35 protein, but the well-visible peripheral ring is not so apparent. One difference is there – they used DMSO treatment (probably due to the fact that tested small-compound must be dissolved in DMSO). Based on this fact, it is hard to compare our data without DMSO with the results of their experiments using DMSO testament for the control samples. However, in general, the cell membrane is abundant on the SC-35. Tripathi and Parnaik (2008) mentioning localization of the substantial pool of GFP-SC-35 in the nascent nuclear envelope of the cells in telophase, which we rather saw in the case of A-type lamin proteins (see revised Fig. 8b). Due to phenomenon, we studied a link between A-type lamins and the SC-35, but there was a low degree of interaction - A-type lamins decorated metaphase chromosomes and well-visible SC-35- positive ring on the cell periphery we detected in telophase (Fig. 8a, b). On the other hand, lamin depletion changed the properties of the SC-35. Tripathi and Parnaik (2008) discussed the existence of a hyperphosphorylated form of SC-35 in MIGs; thus, changes in the phosphorylation status can be responsible for different distribution properties of splicing factor, SC-35. Maybe, DMSO can change phosphorylation – but it is our speculation, not based on experimental data. In the paper of Tripathi and Parnaik (2008), they also studied exogenous protein GFP-SC-35 and SC-35 positive peripheral ring is also barely visible, but similarly, as in Rai et al. (2018), the periphery of the cells in telophase is of a high density of the SC-35 protein. Here, for the revision, we also studied exogenous EGFP- SRSF2, and we observed a high density of this protein in the cytoplasm.

Round 3
Reviewer 2 Report
The manuscript by Legartova and colleagues is acceptable for publish.